# JaxMARL: Multi-Agent RL Environments and Algorithms in JAX

**Alexander Rutherford**[1][*][†]    **Benjamin Ellis**[1][*][†]    **Matteo Gallici**[2][*][†]    **Jonathan Cook**[1][†]
**Andrei Lupu**[1][†]    **Garðar Ingvarsson**[3][†]    **Timon Willi**[1][†]    **Ravi Hammond**[1][†]
**Akbir Khan**[3]    **Christian Schroeder de Witt**[1]    **Alexandra Souly**[3]
**Saptarashmi Bandyopadhyay**[4]    **Mikayel Samvelyan**[3]    **Minqi Jiang**[3]    **Robert Lange**[5]
**Shimon Whiteson**[1]    **Bruno Lacerda**[1]    **Nick Hawes**[1]    **Tim Rocktäschel**[3]
**Chris Lu**[1][*][†]    **Jakob Foerster**[1]
[1]University of Oxford, [2]Universitat Politècnica de Catalunya, [3]University College London,
[4]University of Maryland, [5]Technical University Berlin

## Abstract

Benchmarks are crucial in the development of machine learning algorithms, with available environments significantly influencing reinforcement learning (RL) research. Traditionally, RL environments run on the CPU, which limits their scalability with typical academic compute. However, recent advancements in JAX have enabled the wider use of hardware acceleration, enabling massively parallel RL training pipelines and environments. While this has been successfully applied to single-agent RL, it has not yet been widely adopted for multi-agent scenarios. In this paper, we present JaxMARL, the first open-source, Python-based library that combines GPU-enabled efficiency with support for a large number of commonly used MARL environments and popular baseline algorithms. Our experiments show that, in terms of wall clock time, our JAX-based training pipeline is around 14 times faster than existing approaches, and up to 12500x when multiple training runs are vectorized. This enables efficient and thorough evaluations, potentially alleviating the evaluation crisis in the field. We also introduce and benchmark SMAX, a JAX-based approximate reimplementation of the popular StarCraft Multi-Agent Challenge, which removes the need to run the StarCraft II game engine. This not only enables GPU acceleration, but also provides a more flexible MARL environment, unlocking the potential for self-play, meta-learning, and other future applications in MARL. The code is available at https://github.com/flairox/jaxmarl.

## 1 Introduction

Benchmarks are crucial for developing new single and multi-agent reinforcement learning (MARL) algorithms. They define problems, enable comparisons, and focus research efforts. For example, the development of MuZero was driven by the challenges presented by Go and Chess [54]. Similarly, decentralised StarCraft Micromanagement tasks [18] led to the creation of algorithms like QMIX [51], a popular MARL technique.

In RL research, the runtime of simulations and algorithms is a critical factor affecting the efficiency, thoroughness, and feasibility of experiments. RL training pipelines often require a large number of environment interactions and long, expensive, experimental runs significantly impede research progress. Hardware acceleration and parallelization is an approach to address this: by running

---

[*]Equal Contribution

[†]Core Contributor

38th Conference on Neural Information Processing Systems (NeurIPS 2024) Track on Datasets and Benchmarks.

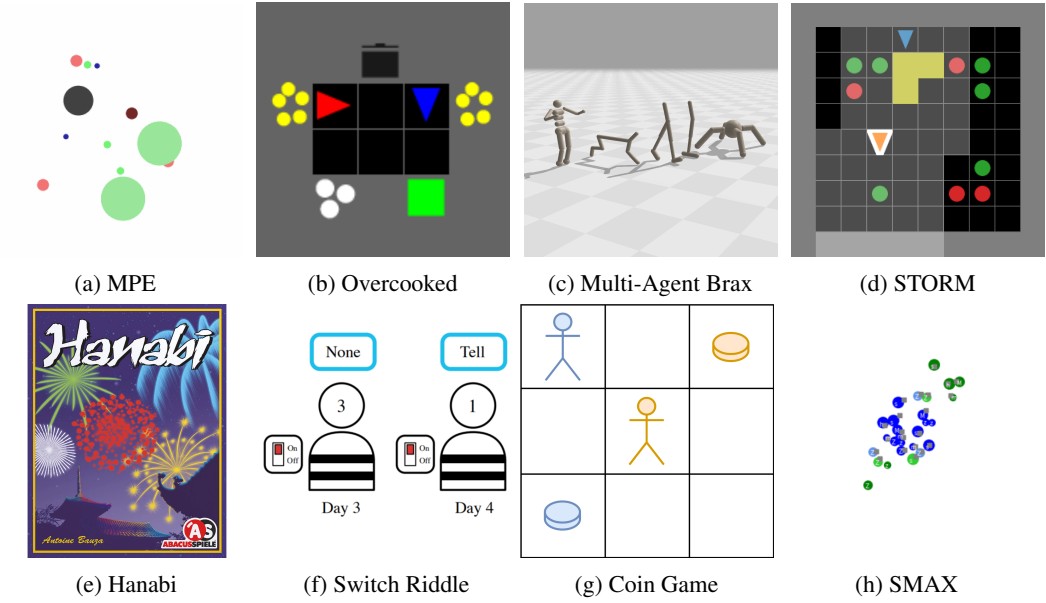

|       |       |       |       |
|-------|-------|-------|-------|
| (a) MPE | (b) Overcooked | (c) Multi-Agent Brax | (d) STORM |
| (e) Hanabi | (f) Switch Riddle | (g) Coin Game | (h) SMAX |

Figure 1: JaxMARL environments. We provide JAX-based implementations of a wide range of customizable MARL environments, covering continuous and discrete dynamics, variable number of agents, full and partial observability, and cooperative, competitive and mixed-incentive settings.

environments on hardware accelerators (e.g. GPUs), we can use many more environment instances in parallel than is feasible with a CPU, drastically improving runtime. However, such an approach typically requires significant engineering effort and often relies on non-Python codebases [56], which reduces its accessibility for many Machine Learning researchers where Python is the lingua franca. That said, recent releases, such as the JAX [7] library and PyTorch's `functorch` module [22], have improved accessibility by enabling Python code to be parallelized and just-in-time compiled on hardware accelerators. This laid the foundation for PureJaxRL [38], which leveraged JAX to implement a parallelized approach, demonstrating that running both the environment and the model training on the same GPU yields a 10x speedup over a traditional pipeline with a GPU-trained policy but a CPU-based environment, and 4000x when multiple training runs are vectorized. This speedup enables new research directions [40, 28] and makes large-scale RL research more accessible [46].

We introduce JaxMARL, which brings these benefits to multi-agent learning. To the best of our knowledge, JaxMARL is the first open-source, Python-based library which leverages JAX for GPU acceleration and supports a wide range of popular MARL environments (as shown in Figure 1) as well as algorithms. We show that MARL greatly benefits from this approach as under the traditional approach, of using CPU-based environments, experiments tend to be particularly slow due to the increased computational burden of training multiple agents simultaneously and the higher sample complexity arising from challenges like non-stationarity and decentralized partial observability. Utilizing an end-to-end JAX-based pipeline for MARL significantly accelerates these experiments, opening up new possibilities for research in this field.

Alongside computational issues, MARL research also struggles with thorough evaluation standards [21]. In particular, MARL papers typically only test on a few domains. Of the 75 recent MARL papers analysed by [21], 50% used only one evaluation environment and a further 30% used only two. While SMAC [53] and MPE [36], the two most used environments, have various tasks or maps, the lack of a standard set raises the risk of biased comparisons and incorrect conclusions. This leads to environment overfitting and unclear progress markers. By alleviating computational constraints, JaxMARL allows for rapid evaluations across a broad set of environments and hence is a powerful tool to address MARL's current evaluation crisis.

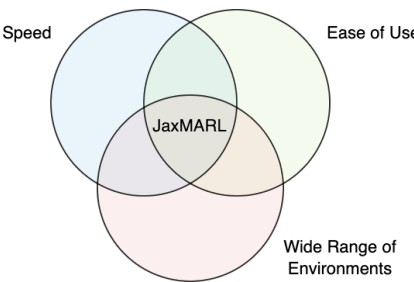

Figure 2: Our philosophy. JaxMARL combines a wide range of environments with ease of use and evaluation speed.

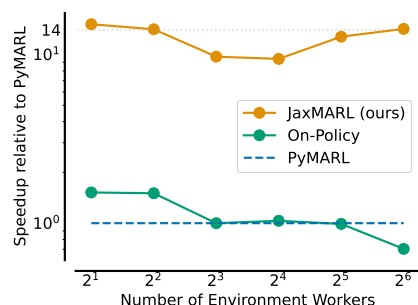

Figure 3: Speed of training an RNN agent using IPPO on a multi-particle environment in JaxMARL compared to two popular MARL libraries, see the Appendix for details.

More specifically, in this paper, our contributions are as follows:

- **JAX Implementations of Popular MARL Environments:** We implement a wide range of popular MARL environments in JAX, enabling fast experimentation across diverse environments. Taking advantage of large scale parallelism, many of our environments run over three orders of magnitude faster on a GPU than their CPU-based counterparts. They are implemented in Python ensuring ease-of-use, following our philosophy set out in Figure 2.

- **New MARL Environment Suites:** We introduce two new MARL environment suites: SMAX and STORM. SMAX is an approximate reimplementation of the popular SMAC(v2) [15] benchmark entirely in JAX, rather than using the StarCraft II game engine like SMAC. It is therefore more customizable and significantly faster: SMAX training is 40,000x faster than the equivalent SMAC implementation on a single NVIDIA 2080 when multiple training runs are vectorized. STORM is a general-sum environment suite inspired by the Melting Pot [34] environment suite that features temporally extended actions within social dilemmas.

- **Implementation of Popular MARL Algorithms in JAX:** We implement many popular MARL algorithms in JAX, such as IPPO, MAPPO and QMIX. As outlined in Figure 3, our training pipeline is up to 14x faster than current popular approaches, and up to 12500x when multiple training runs are vectorized.

- **Comprehensive Benchmarking:** We thoroughly benchmark the speed and correctness of our environments and algorithms, comparing them to existing popular repositories. Generally, our end-to-end JAX implementations run several thousand times faster than their CPU-based counterparts while maintaining equivalent agent performance.

- **Environment Evaluation Recommendations and Best Practice:** Finally, we provide environment evaluation recommendations for different MARL research settings, such as centralized training with decentralized execution and zero-shot coordination. We also provide scripts for large scale evaluation and plotting based on best-practice in the field.

## 2   Background

Our work brings the benefits of hardware acceleration to multi-agent reinforcement learning.

**Hardware Accelerated Environments**   JAX enables the use of Python code with any hardware accelerator, allowing researchers to write hardware-accelerated code easily. Within the RL community, writing environment code in JAX has gained recent popularity. This brings three chief advantages. Firstly, environments written in JAX can be very easily parallelised by using JAX's `vmap` operation, which vectorises a function across an input dimension. Secondly writing the environment in JAX allows the agent and environment to be co-located on the GPU, which eliminates the time taken to copy between CPU and GPU memory. Finally, the code written can be compiled just-in-time, thereby improving performance. Combined, these factors bring significant increases in training speed,

with PureJaxRL [38] achieving a 4000x speedup with vectorized training over traditional training in single-agent settings.

**Multi-Agent Reinforcement Learning Settings**    Multi-Agent Reinforcement Learning is a subfield of reinforcement learning that focuses on environments where multiple agents interact and learn simultaneously. MARL encompasses various settings that address interactions between multiple agents. The most widely-studied MARL setting is the fully-cooperative one, in which agents work together to achieve a common goal. One key framework is centralized training with decentralized execution (CTDE) [37], which allows agents to share information during the learning phase or use additional environment information. During execution, however, agents act based on their independent and partial observations of the environment. Another is zero-shot coordination, which focuses on training agents to coordinate successfully with unseen partners or in new environments without additional training [30]. Beyond fully-cooperative settings, there are zero-sum and general-sum [34] benchmarks that study competitive and mixed incentive interactions.

# 3    JaxMARL

We present JaxMARL, a library containing simple and accessible JAX implementations of popular MARL environments and algorithms. JaxMARL enables significant acceleration and parallelisation over existing implementations. To the best of our knowledge, JaxMARL is the first open-source library that provides JAX-based implementations of a wide range of both MARL environments and baselines. JaxMARL's *interface* is inspired by PettingZoo [61] and Gymnax [32], while the *design philosophy* is based on PureJaxRL [38] and CleanRL [26]. We designed it to be a simple and easy-to-use interface for a wide range of MARL problems. A full specification is provided in the Appendix.

## 3.1    Environments

JaxMARL contains a diverse range of JAX reimplementations of *existing* environments. It also *introduces* SMAX, a novel SMAC-like JAX environment, and STORM, an expansion of matrix games to grid-world scenarios. In this section, we introduce our environments while further details on their implementations can be found in the Appendix.

We measure the speed of our environments in steps per second when using random actions and compare our speed to that of the original environments in Table 3, see the Appendix for details.

### 3.1.1    New Environments

**SMAX**    The StarCraft Multi-Agent Challenge (SMAC) is a popular benchmark in cooperative multi-agent reinforcement learning (MARL) but has several limitations. SMAC's environment lacks sufficient stochasticity for complex policies [15], and its reliance on the StarCraft II engine makes it slow and memory-intensive [43]. Additionally, StarCraft II's constraints limit scenario variety and do not support competitive self-play without significant engineering. To address these issues, we introduce SMAX, a SMAC-like, hardware-accelerated, customizable environment. SMAX features more lightweight dynamics and a less exploitable AI. SMAX incorporates original SMAC scenarios and scenarios similar to those in SMACv2, but is also far more customizable. We provide more details on SMAX and how it improves on SMAC and SMACv2 in the Appendix.

**Spatial-Temporal Representations of Matrix Games (STORM)**    Inspired by the "in the Matrix" games in Melting Pot 2.0 [1], the STORM [29] environment expands on matrix games by representing them as grid-world scenarios. Agents collect resources which define their strategy during interactions and are rewarded based on a pre-specified payoff matrix. STORM can represent cooperative, competitive or general-sum games, like the prisoner's dilemma [57]. Thus, STORM can be used for studying paradigms such as *opponent shaping*, where agents act with the intent to change other agents' learning dynamics, which has been empirically shown to lead to more prosocial outcomes [17, 65, 41, 29, 69]. Compared to the Coin Game or simple matrix games, the grid-world setting presents a variety of new challenges such as partial observability, multi-step agent interactions, temporally-extended actions, and longer time horizons. Unlike the "in the Matrix" games from Melting Pot, STORM features stochasticity, increasing the difficulty [15].

### 3.1.2 Existing Environments

We have also provided JAX-based implementations of several existing environments.

**Hanabi** [3] is a fully-cooperative partially-observable multiplayer card game, where players can observe others' cards but not their own. It is a common benchmark for zero-shot coordination, theory of mind, and ad-hoc teamplay research [23, 24, 9, 40].

**Overcooked** is commonly used for assessing fully-cooperative and fully-observable Human-AI task performance. Our implementation mimics the original from Overcooked-AI [10]. For a discussion on this environment's limitations see [33].

**MABrax** is a derivative of Multi-Agent MuJoCo [49], an extension of the MuJoCo Gym environment [62] that is commonly used for benchmarking continuous multi-agent robotic control.

**Multi-Agent Particle Environment (MPE)** tasks feature a 2D world with simple physics where particle agents can move, communicate, and interact with fixed landmarks [36].

**Coin Game** is a two-player grid-world environment which emulates social dilemmas such as the iterated prisoner's dilemma [57]. While this is a common benchmark for the general-sum setting, previous work [29] has illustrated issues which STORM corrects.

**Switch Riddle** [16] is a simple cooperative communication task included as a debugging tool.

## 3.2 Algorithms

In this section, we present our re-implementation of five well-known MARL baseline algorithms using JAX. All of our training pipelines are fully compatible with JAX's `jit` and `vmap` functions, resulting in significant acceleration of the training processes, as outlined in Figure 3. It also enables parallelisation of training across many seeds and hyperparameters on a single GPU. We follow CleanRL's philosophy of providing clear, single-file implementations [26] and provide a brief overview of the implemented baselines in the Appendix.

**PPO** We implement both Independent PPO (IPPO) [55, 14] and Multi-Agent PPO (MAPPO) [67], with both implementations based on PureJaxRL [38]. We utilise parameter sharing across homogeneous agents and provide both feed-forward and RNN policies.

**Q-learning** Our Q-Learning baselines, including Independent Q-Learning (IQL) [60], Value Decomposition Networks (VDN) [59], and QMIX [52], have been implemented in accordance with the PyMARL codebase [52] to ensure consistency with published results and enable direct comparisons with PyTorch.

## 4 Evaluation Recommendations

Previous work [21] has found significant differences in the evaluation protocols between MARL research works. We identify four main research areas that would benefit from our library: cooperative centralised training with decentralised execution (CTDE) [16], zero-shot coordination [23], general-sum games, and cooperative continuous action methods.

To aid comparisons between methods, we recommend standard *minimal* sets of evaluation environments for each of these settings in Table 1. It's important to note that these are *minimal* and we encourage as broad an evaluation as possible. For example, in the zero-shot coordination setting, all methods should be able to evaluate on Hanabi and Overcooked. However, it may also be possible to evaluate such methods on the SMACv2 settings of SMAX. Similarly, SMAX could be used to evaluate two-player zero-sum methods by training in self-play. For some settings, such as continuous action environments and general-sum games, there is only one difficult environment. We encourage further development of JAX-based environments in these settings to improve the quality of evaluation.

To compute aggregate performance statistics, we follow the recommendations of [2] and evaluate the inter-quartile mean across the different classes of environments. To do this, we recommend normalising performance of the algorithms on the relevant classes of environment (for example via looking at the maximum and minimum performance across algorithms as discussed in [21]), and

Table 1: Recommended minimal environment evaluation sets for different research settings

| Setting | Recommended Environments |
|---|---|
| CTDE | SMAX (all scenarios), Hanabi (2-5 players), Overcooked |
| Zero-shot Coordination | Hanabi (2 players), Overcooked (5 basic scenarios) |
| General-Sum | STORM (iterated prisoner's dilemma), STORM (matching pennies) |
| Cooperative Continuous Actions | MABrax |

computing a mean *per seed*. Then compute the inter-quartile mean across aggregated statistics in each environment. We provide code for performing this calculation. This allows environment classes to be compared fairly, without over-weighting those with more individual scenarios.

## 5 Results

To demonstrate the utility of our library, we evaluate PPO against Q-Learning algorithms on a range of cooperative environments. We discover that not only does it have improved performance, but also that it is more practical to use for end-to-end GPU training.

We then evaluate the speed of our library. We compare algorithm and environment runtimes with similar CPU-based environments. We find that when training PPO, JaxMARL is 31x quicker on SMAX when compared to training in SMAC and 14x quicker on MPE for a single run, and 12,500x for vectorized training runs. Finally, we verify the correctness of our implementations by performing thorough comparisons with prior work.

### 5.1 Multi-Environment Comparison

We provide a preliminary comparison of our PPO and Q-Learning baselines in Figure 4. The IQM and mean were aggregated across 9 SMAX tasks, excluding the two maps with more than 10 units, all 5 Overcooked maps, and the 2 cooperative scenarios of MPE, running 10 seeds per task. We did not evaluate on Hanabi or all SMAX tasks because of the large memory overhead of storing the replay buffer for the Q-Learning methods on the GPU. We normalize the scores of each run on each task against the highest score obtained by any algorithm in that task, and then average the scores in each environment to avoid bias towards SMAX (which contributes more tasks).[1]

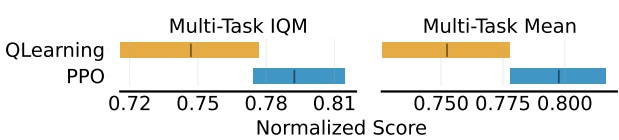

Figure 4: Normalised scores aggregated over SMAX, MPE and Overcooked. PPO shows a clear advantage.

| Environment | PPO | Q-Learning |
|---|---|---|
| SMAX | 10 | 60 |
| Overcooked | 1 | 10 |
| MPE | 1.5 | 2 |
| Hanabi | 450 | - |

Table 2: Mean training time (in minutes) of a single run. PPO is much faster.

The aggregated IQM and Mean scores in Figure 4 show a clear advantage of PPO baselines over Q-Learning. Furthermore, in Table 2 we find that PPO is 6 times faster in SMAX and 10 times faster in Overcooked. We provide additional analysis of this in the Appendix.

### 5.2 Speed Benchmarking

We compare the performance of our environments in steps per second when using random actions to the original environments in Table 3, with details of this test provided in the Appendix.

We next compare the speed of our training pipeline to that of PyMARL. As shown in Figure 5a, a single Q-Learning training run for MPE's simple spread task takes 130 seconds with JaxMARL while PyMARL requires over an hour. Furthermore, using JAX we can parallelise over the entire

---

[1]For PPO, we used MAPPO as the baseline algorithm, except for Overcooked where we used IPPO. For Q-Learning, we used VDN, except for SMAX where we use QMIX as it performs slightly better.

Table 3: Benchmark results for JAX-based MARL environments (steps-per-second) when taking random actions. All environments are significantly faster than existing CPU implementations.

| Environment | Original, 1 Env | Jax, 1 Env | Jax, 100 Envs | Jax, 10k Envs |
|---|---|---|---|---|
| MPE Simple Spread | $8.3 \times 10^4$ | $5.5 \times 10^3$ | $5.2 \times 10^5$ | $4.0 \times 10^7$ |
| Switch Riddle | $2.7 \times 10^4$ | $6.2 \times 10^3$ | $7.9 \times 10^5$ | $6.7 \times 10^7$ |
| Hanabi | $2.1 \times 10^3$ | $1.4 \times 10^3$ | $1.1 \times 10^5$ | $5.0 \times 10^6$ |
| Overcooked | $1.9 \times 10^3$ | $3.6 \times 10^3$ | $3.0 \times 10^5$ | $1.7 \times 10^7$ |
| MABrax Ant 4x2 | $1.8 \times 10^3$ | $2.7 \times 10^2$ | $1.8 \times 10^4$ | $7.6 \times 10^5$ |
| Starcraft 2s3z | $8.3 \times 10^1$ | $5.4 \times 10^2$ | $4.5 \times 10^4$ | $2.7 \times 10^6$ |
| Starcraft 27m vs 30m | $2.7 \times 10^1$ | $1.5 \times 10^2$ | $1.1 \times 10^4$ | $1.9 \times 10^5$ |
| STORM | – | $2.5 \times 10^3$ | $1.8 \times 10^5$ | $1.5 \times 10^7$ |
| Coin Game | $2.0 \times 10^4$ | $4.7 \times 10^3$ | $4.1 \times 10^5$ | $4.0 \times 10^7$ |

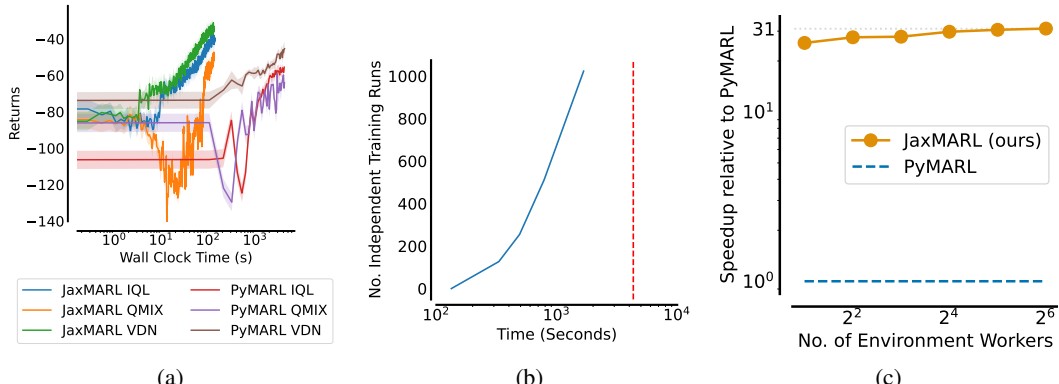

(a)  (b)  (c)

Figure 5: JaxMARL speed benchmarking results. Figure 5a compares JaxMARL's returns in MPE over wall clock time with PyMARL's when using Q-Learning algorithms. Figure 5b demonstrates JaxMARL algorithms' ability to train many seeds in parallel. The figure compares training time (on the x-axis) for a varying number of training runs (on the y-axis) training using QMIX on MPE. The red dotted represents the time taken to train a single agent with PyMARL. Figure 5c illustrates the speedup of a JaxMARL IPPO training run using SMAX compared to PyMARL using SMAC across a varying number of environment rollout threads.

training process within a single hardware accelerator. For QMIX on MPE, this allows us to complete 1024 individual training runs in 198.4 seconds, compared to 1 hour and 10 minutes for a single training run with PyMARL, a speed up of 21,500x per agent. This analysis is repeated for IPPO in the Appendix and we find a speedup of 12,500x. Figure Figure 5c demonstrates the speedup gained from using SMAX with JaxMARL's IPPO implementation compared to training on SMAC with PyMARL. Across a varying number of environment rollout threads, JaxMARL gives a speedup of up to 31x.

## 5.3 Algorithm and Environment Correctness

In this section, we compare our environment and algorithm implementations with prior work and demonstrate equivalence where applicable.

**Overcooked** The transition dynamics of our Overcooked implementation match those of the Overcooked-AI implementation. We demonstrate this by training an IPPO policy on our implementation and evaluating the policy on both our implementation and the original at regular intervals. The performance is similar across the implementations. Results can be found in the Appendix.

**SMAX** SMAX and SMAC are different environments, as they have different opponent policies and dynamics. However, we demonstrate some similarity between them by comparing our IPPO and MAPPO implementations against MAPPO results on SMAC, using the implementation from [58]. We show this figure, along with a more in-depth description of their differences, in the appendix.

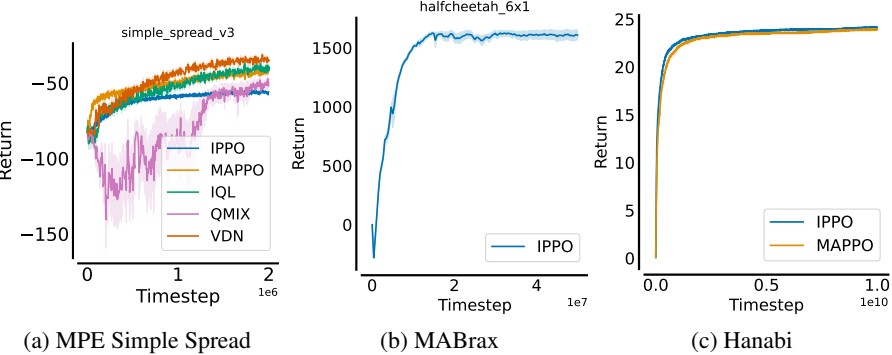

| (a) MPE Simple Spread | (b) MABrax | (c) Hanabi |

Figure 6: Training Curves for a range of JaxMARL environments. Performance is aggregated across 10 seeds and error bars show standard error.

We additionally present aggregate and detailed performance across SMAX in the Appendix. The PPO-based IPPO and MAPPO perform better than the Q-Learning methods, with the centralised information provided in the state helping MAPPO significantly outperform IPPO.

**MPE** Our MPE environment corresponds exactly to the PettingZoo implementation. We validate this for each environment using a uniform-random policy on 1000 rollouts, ensuring all observations and rewards are within a tolerance of $1 \times 10^{-4}$ at each transition. We additionally compare the results of our Q-Learning and PPO implementations with existing libraries, the results of which, along with the performance of IQL on the remaining MPE environments, can be found in the Appendix. We compare Q-Learning and PPO on Simple Spread in Figure 6a.

**MABrax** As Brax differs subtly from MuJoCo, MABrax does not correspond to MAMuJoCo but the learning dynamics are qualitatively similar. Results therefore are not directly comparable across the two environments. We report mean training return across 10 seeds for IPPO on `halfcheetah_6x1` in Figure 6b. We additionally report the training curves for IPPO on `ant_4x2`, `hopper_3x1`, `walker2d_2x3` and `humanoid_9|8` in the Appendix.

**Hanabi** Our implementation matches the Hanabi Learning Environment. To verify the environment's accuracy, we obtained 10,000 action trajectories from the original C++ repository using their pretrained models. We confirmed that processing these action trajectories with JaxMARL produces the same states and returns as the C++ repository. In addition, we transferred the models trained with Pytorch/C++ to Jax and verified they obtain similar scores in JaxMARL. Finally, as shown in Figure 6c, our IPPO and MAPPO models attain scores $\mathbf{24.18 \pm 0.04}$ and $\mathbf{23.95 \pm 0.09}$ respectively, which are better than (for IPPO) or similar to (for MAPPO) the scores attained in [67].

## 6 Related Work

**MARL Libraries and Algorithms** Several open-source libraries exist for both MARL algorithms and environments. The popular library PyMARL [53] provides PyTorch implementations of QMIX, VDN and IQL and integrates easily with SMAC. E-PyMARL [48] extends this by adding the actor-critic algorithms MADDPG [36], MAA2C [44], IA2C [44], and MAPPO, and supports SMAC, Gym [8], Robot Warehouse [11], Level-Based Foraging [11], and MPE environments. Recently released MARLLib [25] is instead based on the open-source RL library RLLib [35] and combines a wide range of competitive, cooperative and mixed environments with a broad set of baseline algorithms. Meanwhile, MALib [70] focuses on population-based MARL across a wide range of environments. However, none of these frameworks feature hardware-accelerated environments and thus lack the associated performance benefits.

**Hardware-Accelerated and JAX-Based RL** There has also been a recent proliferation of hardware-accelerated and JAX-based RL environments. Isaac gym [42] provides a GPU-accelerated simulator for a range of robotics platforms and CuLE [12] is a CUDA reimplementation of the Atari Learning

Environment [4]. Both of these environments are GPU-specific and cannot be extended to other hardware accelerators. Madrona [56] is an extensible game-engine written in C++ that allows for GPU acceleration and parallelisation across environments. However, it requires environment code to be written in C++, limiting its accessibility. VMAS [5] provides a vectorized 2D physics engine written in PyTorch and a set of challenging multi-robot scenarios, including those from the MPE environment. For RL environments implemented in JAX, Jumanji [6] features mostly single-agent environments with a strong focus on combinatorial problems. The authors also provide an actor-critic baseline in addition to random actions. PGX [31] includes several board-game environments written in JAX. Gymnax [32] provides JAX implementations of the BSuite [47], classic continuous control, MinAtar [66] and other assorted environments. Gymnax's sister-library, gymnax-baselines, provides PPO and ES baselines. Further extensions to Gymnax [39] also include POPGym environments [45]. Brax [19] reimplements the MuJoCo simulator in JAX and also provides a PPO implementation as a baseline. Jax-LOB [20] implements a vectorized limit order book as an RL environment that runs on the accelerator. Perhaps the most similar to our work is Mava [50], which provides a MAPPO baseline, as well as integration with the Robot Warehouse environment. None of these libraries combine a range of JAX-based MARL environments with both value-based and actor-critic baselines.

# 7    Conclusion

Hardware acceleration offers important opportunities for MARL research by lowering computational barriers, increasing the speed at which ideas can be iterated, and allowing for more thorough evaluation. We present JaxMARL, an open-source library of popular MARL environments and baseline algorithms implemented in JAX. We combine ease of use with hardware accelerator enabled efficiency to give significant speed-ups compared to traditional CPU-based implementations. Furthermore, by bringing together a wide range of MARL environments under one codebase, we have the potential to help alleviate issues with MARL's evaluation standards. We hope that JaxMARL will help advance MARL by enabling researchers to conduct research with thorough, fast, and effective evaluations.

**Limitations and Future Work.**    While our work provides significant advancements, several limitations remain. First, we observe that the speedups are less pronounced for off-policy, value-based methods. Additionally, there are inherent challenges with end-to-end JAX implementations, such as the difficulty in efficiently handling environments with a variable number of agents or those with massive observation sizes. Furthermore, our MARL environments largely re-implement or draw inspiration from existing environment suites, meaning they do not yet push the boundaries of MARL capabilities. Developing novel MARL environments that push the boundaries of current capabilities could provide new and challenging benchmarks for the community.

# 8 Acknowledgements

This work received funding from the EPSRC Programme Grant "From Sensing to Collaboration" (EP/V000748/1). MG was partially founded by the FPI-UPC Santander Scholarship FPI-UPC_93. JF is partially funded by the UKI grant EP/Y028481/1 (originally selected for funding by the ERC). JF is also supported by the JPMC Research Award and the Amazon Research Award.

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

# Appendix

We structure the appendix as follows. Appendix A provides further background on SMAC, Appendix B describes the environments included within JaxMARL, and Appendix C sets out JaxMARL's API. Appendix D explores JaxMARL's Value-Based MARL methods, including relevant implementation details. Appendix E reports our speed comparisons while Appendix F details training and correctness results not included in the main text. Our hyperparameter values are listed in Appendix G.

# A    Further Background on SMAC

StarCraft is a popular environment for testing RL algorithms. It typically features a centralised controller issuing commands to balance *micromanagement*, the low-level control of individual units, and *macromanagement*, the high level plans for economy and resource management.

SMAC [53], instead, focuses on *decentralised* unit micromanagement across a range of scenarios divided into three broad categories: *symmetric*, where each side has the same units, *asymmetric*, where the enemy team has more units, and *micro-trick*, which are scenarios designed specifically to feature a particular StarCraft micromanagement strategy. SMACv2 [15] demonstrates that open-loop policies can be effective on SMAC and adds additional randomly generated scenarios to rectify SMAC's lack of stochasticity. However, both of these environments rely on running the full game of StarCraft II, which severely increases their CPU and memory requirements. SMAClite [43] attempts to alleviate this computational burden by recreating the SMAC environment primarily in NumPy, with some core components written in C++. While this is much more lightweight than SMAC, it cannot be run on a GPU and therefore cannot be parallelised effectively with typical academic hardware, which commonly has very few CPU cores compared to industry clusters.

# B    Further Details on Environments

## B.1    SMAX

The StarCraft Multi-Agent Challenge (SMAC) is a popular benchmark but has a number of shortcomings. First, as noted and addressed in SMACv2, SMAC is not particularly stochastic. This means that non-trivial win-rates are possible on many SMAC maps by conditioning a policy only on the timestep and agent ID. Additionally, SMAC relies on StarCraft II as a simulator. While this allows SMAC to use the wide range of units, objects and terrain available in StarCraft II, running an entire instance of StarCraft II is slow and memory intensive. StarCraft II runs on the CPU and therefore SMAC's parallelisation is severely limited with typical academic compute.

Using the StarCraft II game engine also constrains environment design. For example, StarCraft II groups units into three races and does not allow units of different races on the same team, limiting the variety of scenarios that can be generated. Secondly, SMAC does not support a competitive self-play setting without significant engineering work. The purpose of SMAX is to address these limitations. It provides access to a simplified SMAC-like, hardware-accelerated, customisable environment that supports self-play and custom unit types. SMAX models units as discs in a continuous 2D space. As listed in Table 4, we include all SMAC(v1) scenarios alongside three inspired by SMAC(v2).

Observations in SMAX are structured similarly to SMAC. Each agent observes the health, previous action, position, weapon cooldown and unit type of all allies and enemies in its sight range. Like SMACv2[15], we use the sight and attack ranges as prescribed by StarCraft II rather than the fixed values used in SMAC.

SMAX and SMAC have different returns. SMAC's reward function, like SMAX's, is split into two parts: one part for depleting enemy health, and another for winning the episode. However, in SMAC, the part which rewards depleting enemy health scales with the number of agents. This is most clearly demonstrated in 27m_vs_30m, where a random policy gets a return of around 10 out of a maximum of 20 because almost all the reward is for depleting enemy health or killing agents, rather than winning the episode. In SMAX, however, 50% of the total return is always for depleting enemy health, and 50% for winning.

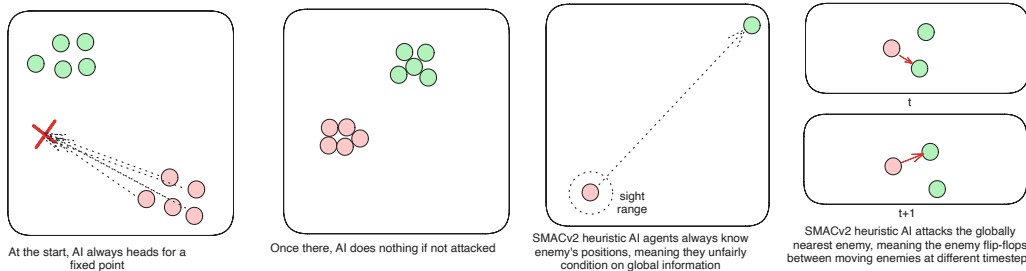

(a) SMAC heuristic AI operation       (b) SMACv2 heuristic AI operation

Figure 7: As shown in Figure 7a, SMAC heuristic AI is decentralised, but does not generalise to new start positions. SMACv2 heuristic AI solves the problem of not being able to locate enemies on the map, but does so via conditioning on the global state, which means that some scenarios might be unwinnable. Additionally, the SMACv2 heuristic AI targets the closest enemy, which can lead to flip-flopping between targets. This is shown in Figure 7b

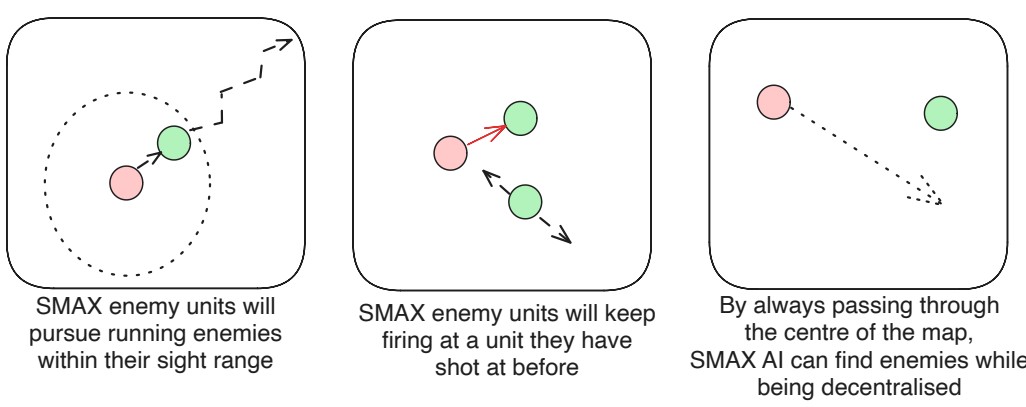

Figure 8: Explanation of the operation of the SMAX heuristic AI.

SMAX also features a different, and more sophisticated, heuristic AI. The heuristic in SMAC simply moves to a fixed location, attacking any enemies it encounters along the way, and the heuristic in SMACv2 globally pursues the nearest agent. Thus the SMAC AI often does not aggressively pursue enemies that run away, and cannot generalise to the SMACv2 start positions, whereas the SMACv2 heuristic AI conditions on global information and is exploitable because of its tendency to flip-flop between two similarly close enemies. SMAC's heuristic AI must be coded in the map editor, which does not provide a simple coding interface. Figure 7 demonstrates these limitations.

In contrast, SMAX features a decentralised heuristic AI that can effectively find enemies without requiring the global information of the SMACv2 heuristic. This guarantees that in principle a 50% win rate is always achievable by copying the decentralised heuristic policy exactly. This means any win-rate below 50% represents a concrete failure to learn. Some of the capabilities of the SMAX heuristic AI are illustrated in the Figure below.

Unlike StarCraft II, where all actions happen in a randomised order in the game loop, some actions in SMAX are simultaneous, meaning draws are possible. In this case both teams get 0 reward.

Like SMAC, each environment step in SMAX consists of eight individual time ticks. SMAX uses a discrete action space, consisting of movement in the four cardinal directions, a stop action, and a shoot action per enemy.

SMAX makes three notable simplifications of the StarCraft II dynamics to reduce complexity. First, zerg units do not regenerate health. This health regeneration is slow at 0.38 health per second, and so likely has little impact on the game. Protoss units also do not have shields. Shields only recharge after 10 seconds out of combat, and therefore are unlikely to recharge during a single micromanagement

task. Protoss units have additional health to compensate for their lost shields. Finally, the available unit types are reduced compared to SMAC. SMAX has no medivac, colossus or baneling units. Each of these unit types has special mechanics that were left out for the sake of simplicity. For the SMACv2 scenarios, the start positions are generated as in SMACv2, with the small difference that the 'surrounded' start positions now treat allies and enemies identically, rather than always spawning allies in the middle of the map. This symmetry guarantees that a 50% win rate is always achievable.

Collisions are handled by moving agents to their desired location first and then pushing them out from one another.

Table 4: SMAX scenarios. The first section corresponds to SMAC scenarios, while the second corresponds to SMACv2.

| Scenario | Ally Units | Enemy Units | Start Positions |
|---|---|---|---|
| 2s3z | 2 stalkers and 3 zealots | 2 stalkers and 3 zealots | Fixed |
| 3s5z | 3 stalkers and 5 zealots | 3 stalkers and 5 zealots | Fixed |
| 5m_vs_6m | 5 marines | 6 marines | Fixed |
| 10m_vs_11m | 10 marines | 11 marines | Fixed |
| 27m_vs_30m | 27 marines | 30 marines | Fixed |
| 3s5z_vs_3s6z | 3 stalkers and 5 zealots | 3 stalkers and 6 zealots | Fixed |
| 3s_vs_5z | 3 stalkers | 5 zealots | Fixed |
| 6h_vs_8z | 6 hydralisks | 8 zealots | Fixed |
| smacv2_5_units | 5 uniformly randomly chosen | 5 uniformly randomly chosen | SMACv2-style |
| smacv2_10_units | 10 uniformly randomly chosen | 10 uniformly randomly chosen | SMACv2-style |
| smacv2_20_units | 20 uniformly randomly chosen | 20 uniformly randomly chosen | SMACv2-style |

## B.2 Spatial-Temporal Representations of Matrix Games (STORM)

This environment features directional agents within an 8x8 grid world with a restricted field of view. For a visual description, see Figure 9. Agents cannot move backwards or share the same location. Collisions are resolved by either giving priority to the stationary agent or randomly if both are moving. Agents collect two unique resources: *cooperate* and *defect* coins. Once an agent picks up any coin, the agent's colour shifts, indicating its readiness to interact. The agents can then release an *interact* beam directly ahead; when this beam intersects with another ready agent, both are rewarded based on the specific matrix game payoff matrix. The agents' coin collections determine their strategies. For instance, if an agent has 1 *cooperate* coin and 3 *defect* coins, there is a 25% likelihood of the agent choosing to cooperate. After an interaction, the two agents involved are frozen for five steps, revealing their coin collections to surrounding agents. After five steps, they respawn in a new location, with their coin count set back to zero. Once an episode concludes, the coin placements are shuffled. This grid-based approach to matrix games can be adapted for n-player versions. While STORM is inspired by MeltingPot 2.0, there are noteworthy differences:

- Meltingpot uses pixel-based observations while we allow for direct grid access.

- Meltingpot's grid size is typically 23x15, while ours is 8x8.

- Meltingpot features walls within its layout, ours does not.

- Our environment introduces stochasticity by shuffling the coin placements, which remain static in Meltingpot.

- Our agents begin with an empty coin inventory, making it easier for them to adopt pure cooperate or defect tactics, unlike in Meltingpot where they start with one of each coin.

- MeltingPot is implemented in Lua [27] where as ours is a vectorized implementation in JAX.

We deem the coin shuffling especially crucial because even large environments representing POMDPs, such as SMAC, can be solved without the need for memory if they lack sufficient randomness [15].

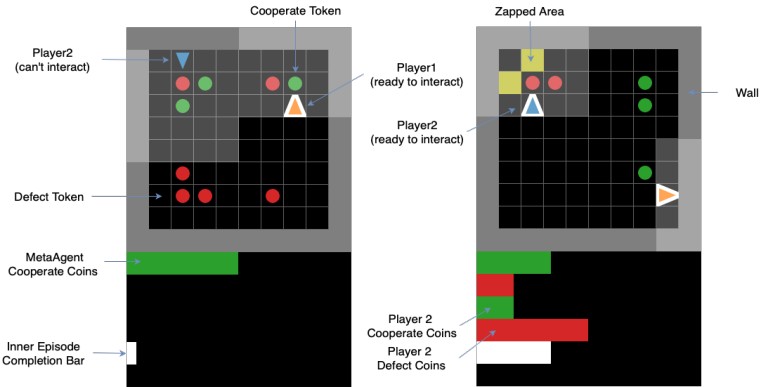

Figure 9: Annotated Image of IPDiTM renders, demonstrating the objects within the game

### B.3 Coin Game

Coin Game is a two-player grid-world environment which emulates social dilemmas such as the iterated prisoner's dilemma [57]. Used as a benchmark for the general-sum setting, it expands on simpler social dilemmas by adding a high-dimensional state. Two players, 'red' and 'blue' move in a grid world and are each awarded 1 point for collecting any coin. However, 'red' loses 2 points if 'blue' collects a red coin and vice versa. Thus, if both agents ignore colour when collecting coins their expected reward is 0.

Two agents, 'red' and 'blue', move in a wrap-around grid and collect red and blue coloured coins. When an agent collects any coin, the agent receives a reward of $1$. However, when 'red' collects a blue coin, 'blue' receives a reward of $-2$ and vice versa. Once a coin is collected, a new coin of the same colour appears at a random location within the grid. If a coin is collected by both agents simultaneously, the coin is duplicated and both agents collect it. Episodes are of a set length.

### B.4 Switch Riddle

Originally used to illustrate the Differentiable Inter-Agent Learning algorithm [16], Switch Riddle is a simple cooperative communication environment that we include as a debugging tool. $n$ prisoners held by a warden can secure their release by collectively ensuring that each has passed through a room with a light bulb and a switch. Each day, a prisoner is chosen at random to enter this room. They have three choices: do nothing, signal to the next prisoner by toggling the light, or inform the warden they think all prisoners have been in the room. The game ends when a prisoner informs the warden or the maximum time steps are reached. The rewards are +1 if the prisoner informs the warden, and all prisoners have been in the room, -1 if the prisoner informs the warden before all prisoners have taken their turn, and 0 otherwise, including when the maximum time steps are reached. We benchmark using the implementation from [68].

### B.5 Hanabi

Hanabi is a fully cooperative partially observable multiplayer card game, where players can observe other players' cards but not their own. To win, the team must play a series of cards in a specific order while sharing only a limited amount of information between players. As reasoning about the beliefs and intentions of other agents is central to performance, it is a common benchmark for ZSC and ad-hoc teamplay research. Our implementation is inspired by the Hanabi Learning Environment [3] and includes custom configurations for varying game settings, such as the number of colours/ranks, number of players, and number of hint tokens. Compared to the Hanabi Learning Environment, which is written in C++ and split over dozens of files, our implementation is a single easy-to-read Python file, which simplifies interfacing with the library and running experiments.

```
1  import jax
2  from jaxmarl import make
3
4  key = jax.random.PRNGKey(0)
5  key, key_reset, key_act, key_step = jax.random.split(key, 4)
6
7  # Initialise and reset the environment.
8  env = make('MPE_simple_world_comm_v3')
9  obs, state = env.reset(key_reset)
10
11 # Sample random actions.
12 key_act = jax.random.split(key_act, env.num_agents)
13 actions = {agent: env.action_space(agent).sample(key_act[i]) \
14            for i, agent in enumerate(env.agents)}
15
16 # Perform the step transition.
17 obs, state, reward, done, infos = env.step(key_step, state, actions)
```

Figure 10: An example of JaxMARL's API, which is flexible and easy-to-use.

## B.6 Overcooked

Inspired by the popular videogame of the same name, Overcooked is commonly used for assessing fully cooperative and fully observable Human-AI task performance. The aim is to quickly prepare and deliver soup, which involves putting three onions in a pot, cooking the soup, and serving it into bowls. Two agents, or *cooks*, must coordinate to effectively divide the tasks to maximise their common reward signal. Our implementation mimics the original from Overcooked-AI [10], including all five original layouts and a simple method for creating additional ones. For a discussion on the limitations of the Overcooked-AI environment, see [33].

## B.7 Multi-Agent Particle Environments (MPE)

The multi-agent particle environments feature a 2D world with simple physics where particle agents can move, communicate, and interact with fixed landmarks. Each specific environment varies the format of the world and the agents' abilities, creating a diverse set of tasks that include both competitive and cooperative settings. We implement all the MPE scenarios featured in the PettingZoo library and the transitions of our implementation map exactly to theirs. We additionally include a fully cooperative predator-prey variant of *simple tag*, presented in [49]. The code is structured to allow for straightforward extensions, enabling further tasks to be added.

## B.8 Multi-Agent Brax (MABrax)

MABrax is a derivative of Multi-Agent MuJoCo [49], an extension of the MuJoCo Gym environment [62] that is commonly used for benchmarking continuous multi-agent robotic control. Our implementation utilises Brax[19] as the underlying physics engine and includes five of *Multi-Agent MuJoCo*'s multi-agent factorisation tasks, where each agent controls a subset of the joints and only observes the local state. The included tasks, illustrated in Figure 1, are: `ant_4x2`, `halfcheetah_6x1`, `hopper_3x1`, `humanoid_9|8`, and `walker2d_2x3`. The task descriptions mirror those from Gymnasium-Robotics [13].

# C JaxMARL's API

The interface of JaxMARL is inspired by PettingZoo [61] and Gymnax. We designed it to be a simple and easy-to-use interface for a wide-range of MARL problems. An example of instantiating an environment from JaxMARL's registry and executing one transition is presented in Figure 10. As JAX's JIT compilation requires pure functions, our `step` method has two additional inputs compared to PettingZoo's. The `state` object stores the environment's internal state and is updated with each call to `step`, before being passed to subsequent calls. Meanwhile, `key_step` is a pseudo-random key, consumed by JAX functions that require stochasticity. This key is separated from the internal state for clarity.

Similar to PettingZoo, the remaining inputs and outputs are dictionaries keyed by agent names, allowing for differing action and observation spaces. However, as JAX's JIT compilation requires arrays to have static shapes, the total number of agents in an environment cannot vary during an

episode. Thus, we do not use PettingZoo's *agent iterator*. Instead, the maximum number of agents is set upon environment instantiation and any agents that terminate before the end of an episode pass dummy actions thereafter. As asynchronous termination is possible, we signal the end of an episode using a special `"__all__"` key within `done`. The same dummy action approach is taken for environments where agents act asynchronously (e.g. turn-based games).

To ensure clarity and reproducibility, we keep strict registration of environments with suffixed version numbers, for example "MPE Simple Spread V3". Whenever JaxMARL environments correspond to existing CPU-based implementations, the version numbers match.

## D    Value-Based MARL Methods and Implementation details

Key features of our framework include parameter sharing, a recurrent neural network (RNN) for agents, an epsilon-greedy exploration strategy with linear decay, a uniform experience replay buffer, and the incorporation of Double Deep Q-Learning (DDQN) [64] techniques to enhance training stability. We stored the replay buffer in GPU memory using Flashbax [63].

Unlike PyMARL, we use the Adam optimizer as the default optimization algorithm. Below is an introduction to common value-based MARL methods.

**IQL** (Independent Q-Learners) is a straightforward adaptation of Deep Q-Learning to multi-agent scenarios. It features multiple Q-Learner agents that operate independently, optimizing their individual returns. This approach follows a decentralized learning and decentralized execution pipeline.

**VDN** (Value Decomposition Networks) extends Q-Learning to multi-agent scenarios with a centralized-learning-decentralized-execution framework. Individual agents approximate their own action's $Q$-Value, which is then summed during training to compute a jointed $Q_{tot}$ for the global state-action pair. Back-propagation of the global DDQN loss in respect to a global team reward optimizes the factorization of the jointed $Q$-Value.

**QMIX** improves upon VDN by relaxing the full factorization requirement. It ensures that a global $argmax$ operation on the total $Q$-Value ($Q_{tot}$) is equivalent to individual $argmax$ operations on each agent's $Q$-Value. This is achieved using a feed-forward neural network as the mixing network, which combines agent network outputs to produce $Q_{tot}$ values. The global DDQN loss is computed using a single shared reward function and is back-propagated through the mixer network to the agents' parameters. Hypernetworks generate the mixing network's weights and biases, ensuring non-negativity using an absolute activation function. These hypernetworks are two-layered multi-layer perceptrons with ReLU non-linearity.

**Issues found when using Q-Learning in an end-to-end GPU setting**. As discussed in the paper's results section, PPO demonstrates a clear advantage over Q-Learning for our benchmarked environments, both in agent performance and training runtime. The speed differential is caused by the optimal sampling/replay ratio for Q-Learning methods becoming rapidly unbalanced as the number of parallel environments increases, which requires us to use fewer parallel environments than we use with PPO. PPO also has a major advantage over Q-Learning in that it does not use a replay buffer, which can occupy a significant amount of GPU memory. Secondly, our experiments empirically showed PPO to be more stable during training.

**A possible workaround** is to increase the replay ratio by performing multiple update steps per training episode, which nevertheless affects computational efficiency. A better solution is to implement a distributed framework, separating the learning and sampling process, which is also out-of-scope for this work.

## E    Speed Comparison

The runs reported in Figures 3 and 5(c) were all run on the same system featuring two NVIDIA GeForce RTX 4090s (although only one was used for training), an Intel(R) Xeon(R) Silver 4316 CPU (20 cores with 40 threads), and 132 GB of RAM. We report the average environment steps per second over the entire RL training process, which for JaxMARL includes any compilation time. For Table 3, all results were collected on a single NVIDIA A100 GPU and AMD EPYC 7763 64-core processor. Environments were rolled out for 1000 sequential steps.

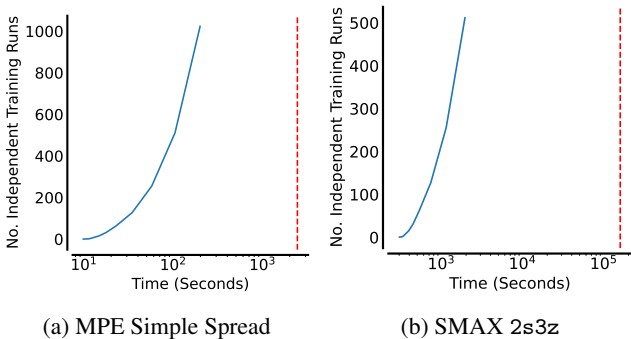

(a) MPE Simple Spread        (b) SMAX 2s3z

Figure 11: Time taken to train a varying number of seeds in parallel on the same device for JaxMARL IPPO (in blue) compared to the time taken to train one seed with MARLLIB (shown as the red dashed line)

In Figure 11 we repeat the analysis, reported in the main paper for QMIX, of JaxMARL's ability to train multiple seeds in parallel for IPPO. Training this way allows training agents many thousands of times faster, with a 12500x speed up in the MPE simple spread environment.

## F    Training & Correctness Results

### F.1    Overcooked

We train IPPO, VDN and IQL agents in Overcooked and present their aggregate performance in Figure 12a. IPPO performs better than the Q-Learning methods in inter-quartile mean and mean, in line with our more general findings. During training, we use the same shaped reward as stated in the original Overooked paper [10], which is added to the score of the game with a factor that is decayed from 1 to 0 during the first half of training. We don't train MAPPO and QMIX for this task because, in Overcooked, agents can observe the entire state of the map. Therefore, there is no partial observability that can be improved through centralized training. We demonstrate correspondence by training an IPPO policy with JaxMARL on our implementation and evaluating the policy over 10 rollouts for both our Overcooked implementation and the original. Results are shown in Figure 14 with the similarity in performance demonstrating their equivalence.

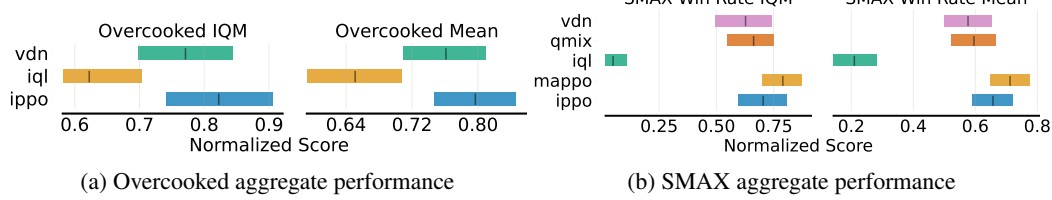

(a) Overcooked aggregate performance       (b) SMAX aggregate performance

Figure 12: Aggregate performance in Overcooked and SMAX for a range of algorithms. Performance is aggregated across 10 seeds and error bars represent 95% bootstrapped confidence intervals as recommended in [2].

### F.2    MABrax

The performance of IPPO on `ant_4x2`, `humanoid_9|8`, `hopper_3x1` and `walker2d_2x3` is reported in Figure 15, with hyperparameters reported in Table 7.

### F.3    MPE

Performance of $Q$-Learning baselines in all the MPE scenarios are reported in Figure 17 and Figure 18. The upper row represents cooperative scenarios, with results for all our $Q$-learning baselines reported.

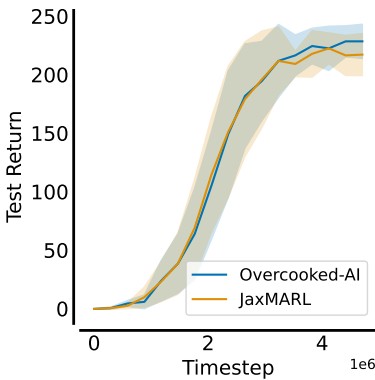

Figure 13: Evaluation performance throughout training of an IPPO policy trained with JaxMARL on our Overcooked Cramped Room scenario implementation and the original [10]. The similarity in performance demonstrates correspondence.

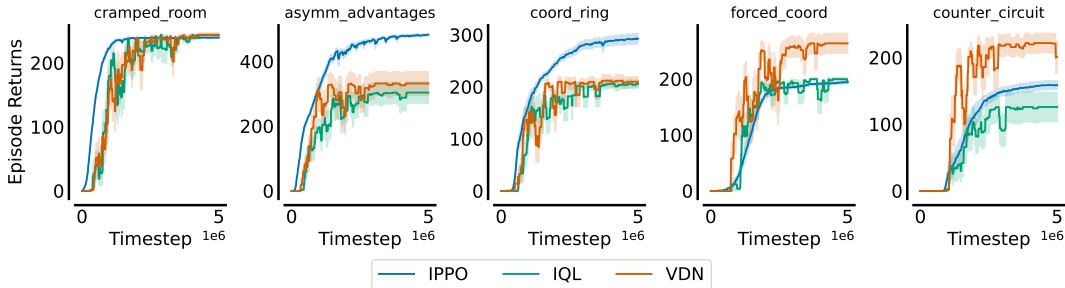

Figure 14: Evaluation of all algorithms in Overcooked scenarios. These scores are obtained after our own hypeparameter tuning, which held to better performances than the using original hyperparameters from Overcooked paper.

The bottom row refers to competitive scenarios, and results for IQL are divided by agent types. Hyperparameters are given in Table 8.

## F.4   SMAX

The performance of different algorithms in SMAX versus MAPPO in SMAC is shown in Figure 19. Hyperparameters for IPPO and the $Q$-learning methods are given in Table 6 and Table 8 respectively. Some maps are significantly more difficult in SMAX, such as `10m_vs_11m`, whereas some are much easier such as `3s_vs_5z`.

## F.5   Hanabi

The performances of our implementation of IPPO are JaxMARL's Hanabi for 2-3 players are reported in Table 5, together with the results provided for IPPO in the original Hanabi environment reported by [67].

| # Players | Metric | IPPO [67] | IPPO (JaxMARL) |
|-----------|--------|-----------|----------------|
| 2 | Avg. | 24.00 | 23.95 |
|   | Best | 24.19 | 24.12 |
| 3 | Avg. | 23.25 | 23.83 |
|   | Best | 23.87 | 24.16 |

Table 5: Best and Average evaluation scores of the original IPPO implementation [67] and IPPO in JaxMARL's Hanabi.

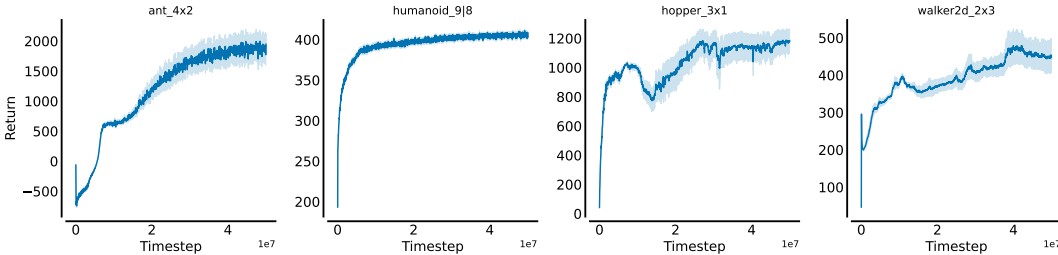

Figure 15: Performance of IPPO on MABrax Tasks

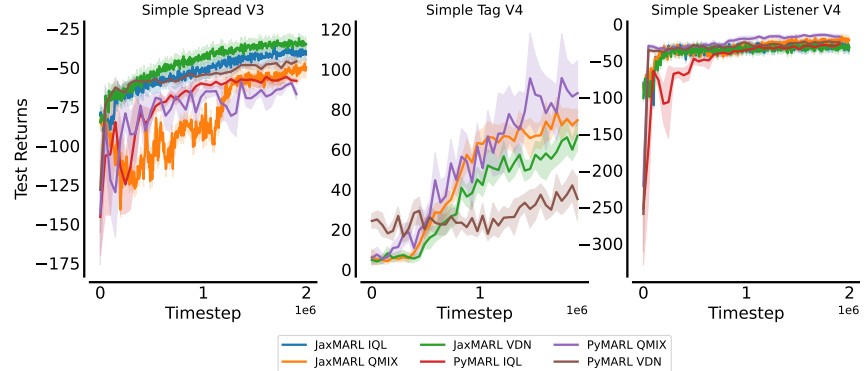

Figure 16: Comparison of the performances of Q-Learning baselines in PyMARL and JaxMARL in two cooperative scenarios of MPE (Spread and Speaker Listener) and one competitive scenario (Simple Tag). For Simple Tag, we pre-trained a prey in JaxMARL and then trained agents to compete with it in both PyMARL and JaxMARL. Despite the small differences in the obtained returns in the two frameworks, the algorithms show similar learning dynamics, and the final ordering is preserved, validating our environment and algorithm implementations.

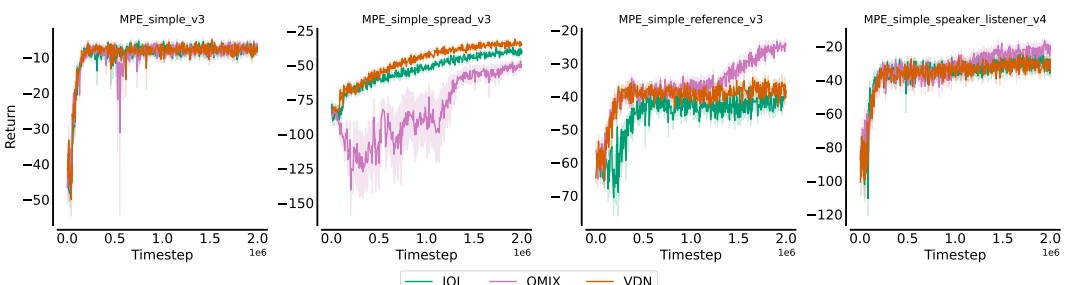

Figure 17: Evaluation of performances of QLearning in all the MPE cooperative scenarios.

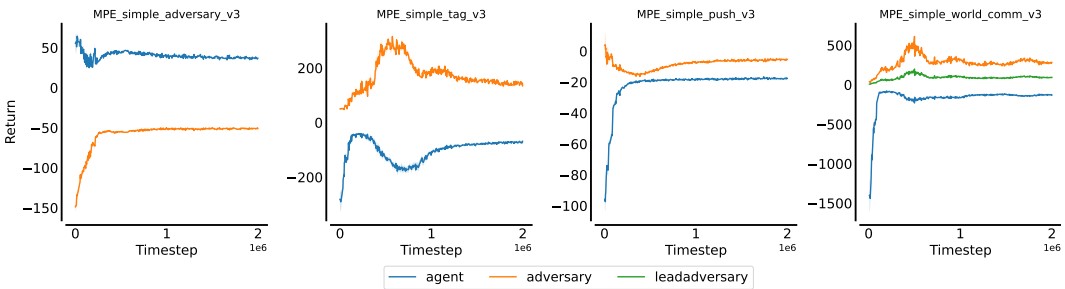

Figure 18: Evaluation of performances of IQL in all the MPE competetive scenarios. All the competetive agents are trained independently together. "Agent" and "Adversary" are teams, not single agents.

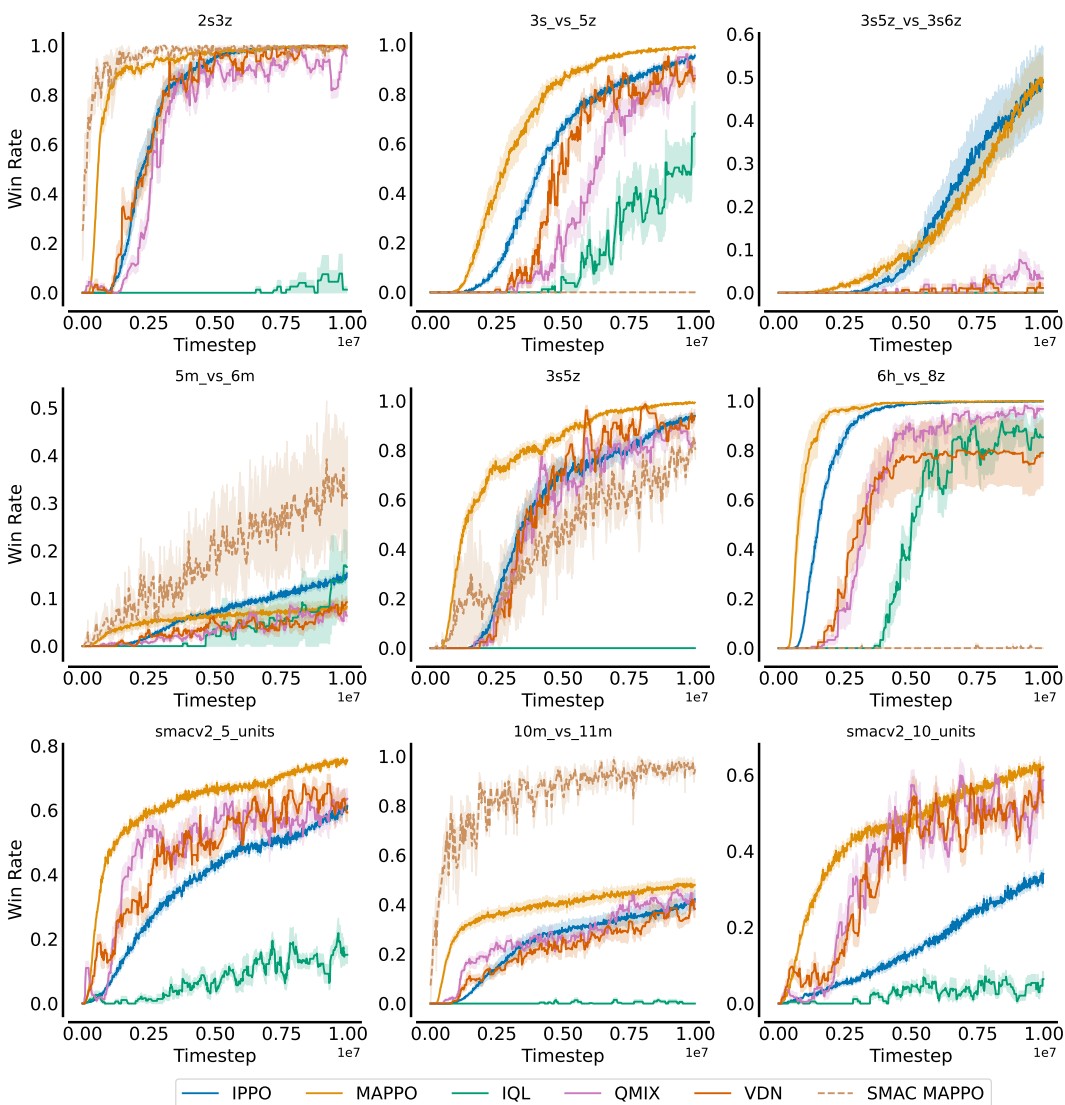

Figure 19: Comparison of IPPO, MAPPO, IQL, QMIX, VDN in SMAX with MAPPO in SMAC.

# G Hyperparameters

Table 6: IPPO hyperparameters for MPE, SMAX, Hanabi and Overcooked.

| Parameter | MPE | SMAX | Hanabi | Overcooked |
|---|---|---|---|---|
| **PPO** | | | | |
| # training timesteps | $1 \times 10^6$ | $1 \times 10^7$ | $1 \times 10^{1}0$ | $5 \times 10^6$ |
| # parallel environments | 16 | 64 | 1024 | 64 |
| # rollout steps | 128 | 128 | 128 | 256 |
| Adam learning rate | $5 \times 10^{-4}$ | $4 \times 10^{-3}$ | $5 \times 10^{-4}$ | $5 \times 10^{-4}$ |
| Anneal learning rate | True | True | True | True |
| Update epochs | 5 | 2 | 4 | 4 |
| Minibatches per epoch | 2 | 2 | 4 | 16 |
| $\gamma$ | 0.99 | 0.99 | 0.99 | 0.99 |
| $\lambda_{\text{GAE}}$ | 1.0 | 0.95 | 0.95 | 0.95 |
| Clip range | 0.3 | 0.2 | 0.2 | 0.2 |
| Entropy coefficient | 0.01 | 0.0 | 0.01 | 0.01 |
| Value loss coefficient | 1.0 | 0.5 | 0.5 | 0.5 |
| Maximum gradient norm | 0.5 | 0.5 | 0.5 | 0.5 |
| Activation | tanh | relu | relu | relu |
| **Feed-forward network** | | | | |
| Number of layers | 2 | - | 2 | 2 |
| Fully-connected width | 64 | - | 512 | 64 |
| **Recurrent network** | | | | |
| Hidden width | 128 | 128 | 128 | - |
| Number of full-connected layers | 2 | 2 | 2 | - |
| Fully-connected width | 128 | 128 | 128 | - |

Table 7: IPPO hyperparameters for MABrax settings.

| Parameter | Ant | HalfCheetah | Walker |
|---|---|---|---|
| **PPO** | | | |
| # training timesteps | $1 \times 10^8$ | | |
| # parallel environments | 64 | | |
| # rollout steps | 300 | | |
| Adam learning rate | $1 \times 10^{-3}$ | $6 \times 10^{-4}$ | $7 \times 10^{-3}$ |
| Anneal learning rate | True | | |
| Update Epochs | 4 | | |
| Minibatches per epoch | 4 | | |
| $\gamma$ | 0.99 | | |
| $\lambda_{\text{GAE}}$ | 1.0 | | |
| Clip range | 0.2 | | |
| Entropy coefficient | $2 \times 10^{-6}$ | $4.5 \times 10^{-3}$ | $1 \times 10^{-3}$ |
| Value loss coefficient | 4.5 | 0.14 | 1.9 |
| Maximum gradient norm | 0.5 | | |
| Activation | tanh | | |
| **Feed-forward network** | | | |
| Number of layers | 2 | | |
| Fully-connected width | 64 | | |

Table 8: Q-Learning hyperparameters for MPE, SMAX and Overcooked

| Parameter | MPE | SMAX | Overcooked |
|---|---|---|---|
| **Q-Learning** | | | |
| # training timesteps | $2 \times 10^6$ | $1 \times 10^7$ | $1 \times 10^5$ |
| # parallel environments | 8 | 16 | 32 |
| # rollout steps | 26 | 128 | 1 |
| Adam learning rate | $1 \times 10^{-3}$ | $5 \times 10^{-5}$ | $7.5 \times 10^{-5}$ |
| Anneal learning rate | True | False | True |
| Buffer size | $5 \times 10^3$ | $5 \times 10^3$ | $1 \times 10^5$ |
| Buffer batch size | 32 | 32 | 128 |
| $\epsilon$ start | 1.0 | 1.0 | 1.0 |
| $\epsilon$ finish | 0.05 | 0.05 | 0.05 |
| $\epsilon$ decay | 0.1 | 0.1 | 0.2 |
| Hidden size | 64 | 512 | 64 |
| Maximum gradient norm | 25 | 10 | 1 |
| $\tau$ | 1.0 | 1.0 | 1.0 |
| Update epochs | 1 | 8 | 4 |
| Learning starts at timestep | $1 \times 10^4$ | $1 \times 10^4$ | $1 \times 10^3$ |
| $\gamma$ | 0.9 | 0.99 | 0.99 |
| **QMIX specific** | | | |
| Mixed embedding width | 32 | 64 | - |
| Mixer hypernet width | 128 | 256 | - |
| Mixer initial scale | $1 \times 10^{-3}$ | $1 \times 10^{-3}$ | - |

