# 7 Hyperparameters

| Value | Ant | HalfCheetah | Walker |
|---|---|---|---|
| VF_COEF | 4.5 | 0.14 | 1.9 |
| ENT_COEF | $2 \times 10^{-6}$ | $4.5 \times 10^{-3}$ | $1 \times 10^{-3}$ |
| LR | $1 \times 10^{-3}$ | $6 \times 10^{-4}$ | $7 \times 10^{-3}$ |
| NUM_ENVS | 64 | – | – |
| NUM_STEPS | 300 | – | – |
| TOTAL_TIMESTEPS | $1 \times 10^{8}$ | – | – |
| NUM_MINIBATCHES | 4 | – | – |
| GAMMA | 0.99 | – | – |
| GAE_LAMBDA | 1.0 | – | – |
| CLIP_EPS | 0.2 | – | – |
| MAX_GRAD_NORM | 0.5 | – | – |
| ACTIVATION | tanh | – | – |
| ANNEAL_LR | True | – | – |

Table 2: MABrax Hyperparameters, where – indicates repeated parameters

| Hyperparameter | Value |
|---|---|
| LR | 0.0005 |
| NUM_ENVS | 25 |
| NUM_STEPS | 128 |
| TOTAL_TIMESTEPS | $1 \times 10^{6}$ |
| UPDATE_EPOCHS | 5 |
| NUM_MINIBATCHES | 2 |
| GAMMA | 0.99 |
| GAE_LAMBDA | 1.0 |
| CLIP_EPS | 0.3 |
| ENT_COEF | 0.01 |
| VF_COEF | 1.0 |
| MAX_GRAD_NORM | 0.5 |
| ACTIVATION | tanh |
| ANNEAL_LR | True |

Table 3: Hyperparameters for MPE IPPO

| Hyperparameter | Value |
|---|---|
| LR | 0.004 |
| NUM_ENVS | 64 |
| NUM_STEPS | 128 |
| TOTAL_TIMESTEPS | $1 \times 10^{7}$ |
| UPDATE_EPOCHS | 2 |
| NUM_MINIBATCHES | 2 |
| GAMMA | 0.99 |
| GAE_LAMBDA | 0.95 |
| CLIP_EPS | 0.2 |
| SCALE_CLIP_EPS | False |
| ENT_COEF | 0.0 |
| VF_COEF | 0.5 |
| MAX_GRAD_NORM | 0.5 |
| ACTIVATION | relu |

Table 4: Hyperparameters for SMAX IPPO

| Hyperparameter | Value |
|---|---|
| LR | $5 \times 10^{-4}$ |
| NUM_ENVS | 1024 |
| NUM_STEPS | 128 |
| TOTAL_TIMESTEPS | $1 \times 10^{10}$ |
| UPDATE_EPOCHS | 4 |
| NUM_MINIBATCHES | 4 |
| GAMMA | 0.99 |
| GAE_LAMBDA | 0.95 |
| CLIP_EPS | 0.2 |
| ENT_COEF | 0.01 |
| VF_COEF | 0.5 |
| MAX_GRAD_NORM | 0.5 |
| ACTIVATION | relu |
| ANNEAL_LR | True |
| NUM_FC_LAYERS | 2 |
| LAYER_WIDTH | 512 |

Table 5: Hyperparameters for Hanabi IPPO

| Hyperparameter | Value |
|---|---|
| LR | 0.0005 |
| NUM_ENVS | 64 |
| NUM_STEPS | 256 |
| TOTAL_TIMESTEPS | $5 \times 10^{6}$ |
| UPDATE_EPOCHS | 4 |
| NUM_MINIBATCHES | 16 |
| GAMMA | 0.99 |
| GAE_LAMBDA | 0.95 |
| CLIP_EPS | 0.2 |
| ENT_COEF | 0.01 |
| VF_COEF | 0.5 |
| MAX_GRAD_NORM | 0.5 |
| ACTIVATION | relu |
| ANNEAL_LR | True |

Table 6: Hyperparameters for Overcooked IPPO

| Hyperparameter | Value | Hyperparameter | Value |
|---|---|---|---|
| NUM_ENVS | 8 | NUM_ENVS | 16 |
| NUM_STEPS | 26 | NUM_STEPS | 128 |
| BUFFER_SIZE | 5000 | BUFFER_SIZE | 5000 |
| BUFFER_BATCH_SIZE | 32 | BUFFER_BATCH_SIZE | 32 |
| TOTAL_TIMESTEPS | $2 \times 10^6$ | TOTAL_TIMESTEPS | $1 \times 10^7$ |
| HIDDEN_SIZE | 64 | HIDDEN_SIZE | 512 |
| MIXER_EMBEDDING_DIM* | 32 | MIXER_EMBEDDING_DIM* | 64 |
| MIXER_HYPERNET_HIDDEN_DIM* | 128 | MIXER_HYPERNET_HIDDEN_DIM* | 256 |
| MIXER_INIT_SCALE* | 0.001 | MIXER_INIT_SCALE* | 0.001 |
| EPS_START | 1.0 | EPS_START | 1.0 |
| EPS_FINISH | 0.05 | EPS_FINISH | 0.05 |
| EPS_DECAY | 0.1 | EPS_DECAY | 0.1 |
| MAX_GRAD_NORM | 25 | MAX_GRAD_NORM | 10 |
| TARGET_UPDATE_INTERVAL | 200 | TARGET_UPDATE_INTERVAL | 10 |
| TAU | 1.0 | TAU | 1.0 |
| NUM_MINI_EPOCHS | 1 | NUM_MINI_EPOCHS | 8 |
| LR | 0.005 | LR | 0.00005 |
| LEARNING_STARTS | 10000 | LEARNING_STARTS | 10000 |
| LR_LINEAR_DECAY | True | LR_LINEAR_DECAY | False |
| GAMMA | 0.9 | GAMMA | 0.99 |

Table 7: QLearning Hyperparameters in MPE (* Hyperparameters specific to QMix.)

Table 8: QLearning Hyperparameters in Smax (* Parameters specific to QMix.)

| Hyperparameter | Value |
|---|---|
| NUM_ENVS | 32 |
| NUM_STEPS | 1 |
| BUFFER_SIZE | $1 \times 10^5$ |
| BUFFER_BATCH_SIZE | 128 |
| TOTAL_TIMESTEPS | $5 \times 10^6$ |
| HIDDEN_SIZE | 64 |
| EPS_START | 1.0 |
| EPS_FINISH | 0.05 |
| EPS_DECAY | 0.2 |
| MAX_GRAD_NORM | 1 |
| TARGET_UPDATE_INTERVAL | 10 |
| TAU | 1.0 |
| NUM_MINI_EPOCHS | 4 |
| LR | 0.000075 |
| LEARNING_STARTS | 1000 |
| LR_LINEAR_DECAY | True |
| GAMMA | 0.99 |

Table 9: QLearning Hyperparameters in Overcooked