# OpenReview forum: "JaxMARL: Multi-Agent RL Environments and Algorithms in JAX"
_NeurIPS.cc/2024/Datasets_and_Benchmarks_Track — NeurIPS 2024 Track Datasets and Benchmarks Poster_

### Official Review · Reviewer_Vp7t · 2024-07-12
**Offical Review for JaxMARL**

**Rating:** 6
**Confidence:** 3
**Clarity:** Yes. The paper is well-written and ea…

**Review:**

quality: good

originality: excellent

significance: fair

Please check the following sections for my detailed review.

**Strengths:**

+ The paper is well-written and easy to follow.

+ The implementation and testing efforts for complex environments like SMAC and Hanabi in JAX are substantial. The authors have done an excellent job with the implementation.

+ The experiment results in the paper are extensive. The speedup with JAX is impressive.

**Additional Feedback:**

No

**Correctness:**

As validated by the unit tests and the learning curve, I believe the implementation is correct.

**Documentation:**

After I checked the release repo, I believe reproducibility is ensured. However, I didn't see complete guides for environment and algorithm customization, which may impede users' usage.

**Limitations:**

The authors have mentioned several limitations in the final section:

+ A fairly low speedup for off-policy algorithms

+ The lack of environment implementation with a variable number of agents (flexibility, weakness#2) and massive observation sizes (distributed training, weakness#4)

+ MARL environments are largely re-implementations of existing ones (weakness#1)

These limitations largely correspond to my personal perception. I have presented my detailed explanations in the weaknesses section.

**Opportunities For Improvement:**

+ While these new environments can help researchers quickly iterate on ideas and adjust hyper-parameters, the paper doesn't offer insights into developing new MARL algorithms or designing MARL environments.

+ The flexibility of MARL environments in JAX is quite limited. As the authors mentioned in the final section, we cannot vary the number of agents in each episode. Since the dataflow is JIT-compiled and fixed, it may also be difficult to implement dynamic policy selection and evolution, such as league training in AlphaStar.

+ When I checked the repository, I didn't find a practical guide for developing a customized environment. Is it easy to do so? If not, researchers might quickly overfit to the existing environments.

+ It's not well justified why Q-learning could not be evaluated in Hanabi and some SMAX environments, given that JAX facilitates storing the replay buffer in multiple GPUs. Another solution is to store the buffer in CPU memory and use it eagerly during training. If the environment itself is very memory-intensive, which can be unavoidable for customized applications, does JaxMARL support multi-GPU training with jax.pmap?

Minor:

+ Some captions are incorrect, e.g., Figure 2 in the text actually refers to Table 2.

Questions:

+ Compared with PureJaxRL, are there any significant difficulties when transitioning from single-agent cases to multi-agent cases?

**Relation To Prior Work:**

The paper has extensively discussed the relationship between prior works. I have listed my perceived contributions in the above section.

**Summary And Contributions:**

This paper presents a collection of multi-agent RL environments implemented in JAX, ranging from simple matrix-based benchmarks to complex environments like SMAC and Hanabi. The authors also implemented popular MARL algorithms, QMIX and PPO, and evaluated them on these benchmarks. They verified the correctness of their environments and algorithm implementations by comparing the JAX-based environment with the plain CPU version and by presenting learning curves. The JaxMARL training pipeline achieves a significant speedup over its CPU counterpart.

The primary contribution comes from the implementation and testing efforts for a wide range of JAX-based MARL environments, especially for complex ones like SMAC and Hanabi. The authors have done an excellent job with the implementation.

While the authors also provide an evaluation of QMIX and PPO in these environments, the benchmark results mainly serve as a certification of the correctness. They do not provide additional insights into developing new MARL algorithms or designing MARL environments.

---

> ### Author Rebuttal · Authors · 2024-08-18
>
> We thank the reviewer for their comments, especially for highlighting our "excellent" effort on implementations which result in "impressive" speedups underlined by "extensive" experimental results proving correctness. We address the issues raised in your review below:
>
> ## Insights into developing new MARL algorithms or environments.
>
> While we agree that new MARL algorithms and environments are crucial, given JaxMARL's substantial contributions we have left this for future work.
>
> ## Flexibility of MARL environments in JAX
> > ...we cannot vary the number of agents in each episode.
>
> The number of agents within an episode can be varied; however, due to JAX's JIT compilation, the *maximum number* possible of agents within an episode must be specified when the environment is instantiated. The range of environments within JaxMARL also illustrates the flexibility of JAX for MARL settings.
>
> > Since the dataflow is JIT-compiled and fixed, it may also be difficult to implement dynamic policy selection and evolution, such as league training in AlphaStar.
>
> Evolution is possible in JAX, as evidenced by the popular `evojax` and `evosax` libraries. Furthermore, Deepmind's AlphaStar implementation (https://github.com/google-deepmind/alphastar) already leverages JAX.
>
> ## Is it easy to add environments?
>
> A customised/new environment need only comply with JaxMARL's API to be fully usable with the suite of baselines. The range of environments currently within JaxMARL demonstrates the generality of our API in allowing any MARL setting to be expressed. Indeed, during the development of the library, several groups worked independently on their environments using only the API as a reference. As such, we are confident that researchers with a working knowledge of JAX can easily extend our library for their needs.
>
> ## Risk of researchers overfitting to existing environments
>
> As raised in our global response, recent research [1] has outlined how many recent MARL works only evaluate on one, or a small number of environments. As such, rather than risking researchers overfitting JaxMARL's fast implementations of a wide range of environments hold the potential to help improve evaluation standards within the field. Furthermore, due to our accessible API, new environments are easy to add as the field progresses.
>
> ## Q-Learning on Hanabi and SMAX
>
> It is certainly possible to store data in local buffers for each GPU. The replay buffer can also be stored on the CPU, which we believe is ideal for memory-intensive applications. However, our baselines are based on end-to-end single-GPU setups and function correctly in most scenarios, so we haven't explored these options yet. Additionally, recent research [2] has shown that JaxMARL environment vectorisation can completely replace the replay buffer in QLearning, offering an even simpler way to run VDN on GPUs also for memory-intensive applications like Hanabi.
>
> ## Does JaxMARL support multi-GPU training with jax.pmap?
>
> Yes, in several ways:
>
> - Our training scripts are end-to-end compilable and can be run in a vectorised manner using `vmap` and `pmap`, enabling multiple seeds to run in parallel on one or multiple devices.
> - Interaction with multiple environments can be parallelised across multiple devices by applying `pmap` to the `reset` and `step` functions.
> - Gradient updates during PPO training can also be parallelised across multiple devices.
>
> For more details, see MAVA (https://github.com/instadeepai/Mava/tree/develop/mava), which integrates `pmap` with JaxMARL.
>
> ## Difficulties in transitioning from single-agent cases to multi-agent cases
>
> We did not encounter any major difficulties when transitioning to the multi-agent case. With parameter sharing (as in our setup), a single network can manage interactions between environments and gradient updates, making the transition straightforward. We only needed to handle interactions with multiple agents as an additional batch dimension
>
> ## Guides for environment and algorithm customization
>
> As mentioned in our global response, JaxMARL is readily customisable. Rather than providing a guide we have focused on ensuring the code is clear and well-documented. This ensures accessibility for any intended MARL use case.
>
>
> ## Conclusion
> We hope we have addressed the reviewer's comments and are happy to discuss any of these points further. We would further ask if our responses have addressed the reviewer's concerns, and that they consider increasing their support for our paper.
>
> ---
> [1] Gorsane, et al. "Towards a Standardised Performance Evaluation Protocol for Cooperative MARL." Advances in Neural Information Processing Systems 35 (2022): 5510-5521.
>
> [2] Gallici, Fellows, et al. "Simplifying Deep Temporal Difference Learning." arXiv preprint arXiv:2407.04811 (2024).

---

> > ### Comment · Reviewer_Vp7t · 2024-08-21
> > **I appreciate the authors' responses.**
> >
> > Thank you for your detailed explanation. I don't have any additional conerns and will remain my original score.

---

> > > ### Author Response · Authors · 2024-08-22
> > > **Response to Vp7t**
> > >
> > > Dear Reviewer,
> > >
> > > Thank you for your prompt response. If you no longer have any concerns, we were curious why our work remains only “marginally above the acceptance threshold.”

---

> > > > ### Author Response · Authors · 2024-08-28
> > > > **Response to Vp7t**
> > > >
> > > > Dear Reviewer, we thank you again for your response. With the discussion period nearing its end, please could you outline why our work remains only “marginally above the acceptance threshold” if you have no additional concerns.

---

> > > > > ### Comment · Reviewer_Vp7t · 2024-08-29
> > > > >
> > > > > Hi Authors,
> > > > >
> > > > > I have decided to maintain my borderline score for the following reasons:
> > > > >
> > > > > 1. As you confirmed in your response, there are no significant challenges in transitioning from the single-agent case to the multi-agent case. Consequently, the novelty of JaxMARL compared to single-agent libraries like PureJaxRL is somewhat limited. Additionally, the provided environments are not new; they primarily reference the original Python/C++ implementations, with the main task being the translation of this code into JAX.
> > > > >
> > > > > 2. It is well-known that PPO generally achieves decent performance in most MARL environments, particularly in established ones like SMAC and Hanabi. Given that the MARL community is currently less focused on developing new algorithms, users might be more interested in how to accelerate their custom environments using PPO, rather than re-running existing algorithms and environments for benchmarking purposes. However, since JaxMARL consists of manual implementations rather than a compiler, it may not be particularly useful in such cases.
> > > > >
> > > > > While I respect the authors' efforts in implementing this library, I believe JaxMARL represents more of a labor-intensive project than a technical innovation or an insightful benchmark that could drive further innovation. Although I think JaxMARL is worthy of publication, e.g., as a tutorial tool for beginners, I remain conservative about its potential for broader contributions.

---

> > ### Author Response · Authors · 2024-08-29
> > **Response to Reviewer Vp7t**
> >
> > Dear Reviewer,
> >
> > Thank you for your response and your continued engagement.
> >
> > On your first point:
> > As mentioned in our Global Response, two of JaxMARL's environments (SMAX and STORM) are novel. While the other environments are indeed reimplementations of existing environments, this does not lessen the contribution of SMAX and STORM. SMAX, in particular, addresses fundamental issues with the most popular MARL benchmarking environment [1], alongside offering a 31x speedup for training pipelines. This is a significant and novel contribution.
> >
> > We respectfully disagree with your claims about the novelty of our paper. We provide a set of high-quality, high-speed, and wide-ranging environments and algorithms. This has not been assembled before in MARL and we believe it will significantly impact the pace and type of research performed. There is significant value in this type of work, which is likely to shape the experimental practice of MARL researchers for a significant time. Consider, for example, the impact of PyMARL, the fragments of which could be found in the codebases of projects many years after its inception.
> >
> >
> > On your second point, while we agree that it is well-known that PPO generally performs well across many MARL environments, there is certainly still important work to be done on developing new algorithms. For example, in Hanabi, PPO fails to achieve perfect performance in self-play and cannot be used to cooperate with humans. As such, we would strongly contest that algorithm development is not of interest to the MARL community.
> >
> > For researchers looking at new environments rather than algorithms, JaxMARL providing high-quality algorithm implementations ensures it is straightforward to leverage JAX’s speedup for any research direction.
> >
> > On JaxMARL not being an "insightful benchmark that could drive further innovation". As stated in our Global Response, recent work [1] has established "that many MARL works only evaluate on one, or a small number of environments. This causes methods to overfit to a small number of environments (primarily SMAC and MPE). As JaxMARL provides fast implementations of a wide range of environments within a single easy-to-use repository, we hope we will help alleviate the evaluation issues within the field." We believe the reviewer has not fully considered this argument as JaxMARL holds the potential to address a fundamental issue in MARL. As illustrated by [1, 2], MARL cannot productively innovate without thorough benchmarking as performance improvements are unclear.
> >
> > Thank you again for your continued engagement and we look forward to hearing your response.
> >
> > ---
> > [1] Gorsane, Rihab, et al. "Towards a standardised performance evaluation protocol for cooperative marl." Advances in Neural Information Processing Systems 35 (2022): 5510-5521.
> >
> > [2] Ellis, Benjamin, et al. "Smacv2: An improved benchmark for cooperative multi-agent reinforcement learning." Advances in Neural Information Processing Systems 36 (2024).

---

> > > ### Comment · Reviewer_Vp7t · 2024-08-30
> > >
> > > Dear Authors,
> > >
> > > Thank you for your additional comments. While I would like to further share my thoughts, I want to emphasize that I acknowledge the contributions of this paper and still vote for acceptance.
> > >
> > > ---
> > >
> > > > SMAX and STORM are novel
> > >
> > > To some extent, yes, as these environments did not exist in previous implementations. However, the core ideas behind these environments (i.e., state/action spaces and dynamics) have long been established and adopted by the community. I believe the primary contribution of JaxMARL lies in its use of JAX to accelerate these environments through necessary re-implementations. This does not mean that the test suites themselves are novel. By "novel" test suites, I refer to environments that offer unique features previously unexplored, such as Hanabi, which was the first turn-based cooperative MARL environment at the time it was introduced. Thus, I don't consider the inclusion of SMAX and STORM in JaxMARL to be a standout feature beyond what I see as the true contribution.
> > >
> > > > offering a 31x speedup for training pipelines
> > >
> > > I am indeed impressed by the speedup, which is why I voted for acceptance.
> > >
> > > > the novelty of our paper.... This has not been assembled before...
> > >
> > > I agree that JAX-based MARL environments were not previously available in any libraries, and the introduction of JaxMARL indeed addresses several existing challenges. However, in my view, "has not been assembled before" signifies a strong "contribution" rather than significant "novelty." JaxMARL extends the concept of a fully JAX-based RL pipeline from PureJaxRL to multi-agent environments. In addition, these environments themselves are not particularly novel, so I do not perceive a major novelty from my perspective.
> > >
> > > > in Hanabi, PPO fails to achieve perfect performance in self-play … we would strongly contest that algorithm development is not of interest to the MARL community
> > >
> > > Thank you for clarifying this point. I agree with the authors' position.
> > >
> > > > JaxMARL providing high-quality algorithm implementations
> > >
> > > While I appreciate your reference implementation, I must note that it primarily involves code translation, which I don't see as a significant contribution or novelty. It is much more straightforward to implement JAX-based RL algorithms than developing environments and completing JAX pipelines. There are countless JAX-based re-implementations of existing PyTorch algorithms on GitHub, but most are not of sufficient quality to be published in a top-tier conference like NeurIPS.
> > >
> > > > JaxMARL holds the potential to address a fundamental issue in MARL
> > >
> > > I agree that JaxMARL has the potential to enhance the **solidness and robustness of new MARL algorithms** in the future, but that is different from "driving further innovations." For example, Hanabi's introduction inspired the concepts of cross-play and zero-shot coordination, leading to many practically valuable algorithms that followed these ideas. I don't believe JaxMARL has the capacity to provide such insights as Hanabi did.

---

> > > > ### Author Response · Authors · 2024-08-31
> > > > **Response to Vp7t**
> > > >
> > > > Dear Reviewer,
> > > >
> > > > Thank you again for your continued engagement and commitment to our work.
> > > >
> > > > As an overall comment, we strongly believe that JaxMARL represents a significant improvement over current MARL frameworks. As the reviewer agrees, we bring impressive speedups for a wide range of environments which have not been assembled before. These speed-ups allow researchers to iterate faster and open new avenues of research while the breath of environments ensures evaluation remains robust. This is a strong contribution that aligns with the Datasets & Benchmark call for papers, which stresses in its first line that "new datasets" alongside "thoughtfully designed (collections of) datasets based on previously available data" are welcome. JaxMARL not only presents new environments but also a collection of existing ones that is "thoughtfully designed" due to the impressive speedups and the wide range of environments ensuring robust evaluation.
> > > >
> > > > We agree with the reviewer that the core ideas of these environments have previously been explored. However, the significance of correcting fundamental issues with popular benchmarks should not be overlooked. While new paradigms are valuable, so are significant improvements to our current tools. This ties into your point on solidness and robustness as this is key to ensure proper progress markers. JaxMARL's speed and range of environments makes robust evaluation much easier and so will help to ensure progress can be properly evaluated. While less *flashy* than a novel environment, we address issues at the heart of the current MARL field.
> > > >
> > > > On the algorithm implementations, we understand the reviewer's position.
> > > >
> > > > Finally, on driving innovations, we would strongly contest that JaxMARL does not have the potential to do this. In fact, and as mentioned in our global response, observations on how our Q-Learning baselines performed in a GPU setting inspired a significant project led by one of the lead authors of our paper [1]. Their work proposes PQN, a Q-learning algorithm suited for use with JAX-based environments that significantly outperforms IPPO/MAPPO on many tasks, including SMAX. This is a clear example of JaxMARL driving further innovations.
> > > >
> > > > Thank you again for your commitment to the discussion period. We appreciate that you recognise the contributions of our paper and hope you will consider increasing your support.
> > > >
> > > > ---
> > > >
> > > > [1] Gallici, Matteo, et al. "Simplifying Deep Temporal Difference Learning." arXiv preprint arXiv:2407.04811 (2024).

---

### Official Review · Reviewer_DMjY · 2024-07-23
**A good contribution with room for improvement in comparing SMAX and SMAC**

**Rating:** 6
**Confidence:** 3
**Clarity:** The paper is well-organized and easy …

**Review:**

The speed advantage of the JAX-implemented environment, as symbolized in Figure 3, is evidenced in Table 3. The learning speed, as shown in Table 2, supports the claim of aiding MARL research.

However, there is a room for improvement as outlined in the "Opportunities For Improvement" section.

**Strengths:**

* The environment is extremely fast, with PPO experiments, excluding Hanabi, completing in about 10 minutes, which is very appealing.
* The environment is diverse and categorized in a way that aligns with the context of MARL research, which is beneficial for researchers.
* Several important benchmark methods are provided, offering a good starting point for MARL researchers.

**Additional Feedback:**

Minor comments:

* Appendix L2: `It typically features features a centralized 3 controller`: repeated `features`.
* References could be improved. For example, some arXiv papers are already published (e.g., Marllib in JMLR, Bsuite in ICLR2020, Pgx in NeurIPS2023, Jumanji in ICLR2024).
* While a Dockerfile is provided, the dependency specifications in `requirements.txt` are extremely strict. To extend the library's lifespan, consider enhancing it to work with fewer dependency constraints.

**Correctness:**

The claim that JaxMARL covers several important benchmarks in MARL research, offers a speed advantage over existing implementations, and provides researchers with low-cost research opportunities is well-supported.

**Documentation:**

The documentation is adequately hosted in the README with sufficient information. The implementation, including benchmarks, is publicly available, ensuring reproducibility.

**Ethics:**

I found no ethics concerns.

**Limitations:**

The limitations are appropriately mentioned in the Conclusion.

**Opportunities For Improvement:**

I believe one of the main contributions of this paper is SMAX, a simplified version of SMAC. The paper could be improved by discussing quantitatively whether SMAX truly inherits the key features of SMAC as a MARL environment. For example, are methods effective in SMAC also useful in SMAX? Can insights gained in SMAX be applied to SMAC?

**Relation To Prior Work:**

The relationship to prior research is appropriately mentioned.

**Summary And Contributions:**

This paper introduces JaxMARL, a JAX-based implementation of benchmarks in MARL, including a simplified version of SMAC called SMAX. It offers researchers a significantly faster research cycle. The experimental results demonstrate the speed advantages of JaxMARL, highlighting its potential to enhance MARL research by providing benchmark results.

---

> ### Author Rebuttal · Authors · 2024-08-18
>
> Dear Reviewer, thank you for your positive review and for noting JaxMARL's speed, diversity of environments and provision of important benchmark methods. Our response to your comments is below.
>
> ## SMAC vs SMAX
>
> As outlined in our global response and our Appendix, **SMAX is an updated, not simplified, version of SMAC.** As a result, performance between SMAC and SMAC scenarios is not directly comparable. This is evidenced in Figure 13 of our Appendix where we compare the performance of JaxMARL baselines in SMAX with `on-policy`'s MAPPO implementation in SMAC. While the baseline methods perform similarly on scenarios, there is generally no correlation; given SMAX's updated enemy policy this is not surprising. However, they still represent similar tasks and such insights gained in SMAX can likely be applied to SMAC and SMACv2.
>
>
> ## Minor Comments
>
> Thank you for your comment on referencing and the typo, we have corrected our manuscript.
>
> We agree that the requirements.txt are restrictive, but we have set them as such to ensure all results are collected within the same Python environment to ensure they are comparable.
>
> ## Conclusion
> We hope we have addressed the reviewer's comments and are happy to discuss any of these points further. We would ask that if our responses have addressed the reviewer's concerns, they consider increasing their support for our paper.

---

> > ### Author Response · Authors · 2024-08-28
> > **Response DMjY**
> >
> > Dear Reviewer, we hope you have had time to review our response. With the discussion period nearing its end, please could you let us know whether we have addressed your concerns or if you still have any questions.

---

> ### Comment · Reviewer_DMjY · 2024-08-30
>
> Thank you for the rebuttal. I have no further concerns.
>
> The clarification regarding the differences and intentions between SMAX and SMAC was convincing. However, I believe SMAC's popularity as a benchmark is partly due to the popularity of StarCraft II as a game. I am uncertain whether SMAX will be widely accepted in the MARL community if it has differences from SMAC (even if reasonable) and does not offer visually appealing demonstrations similar to StarCraft II replays.
>
> I think JaxMARL is an effort worthy of publication, but due to uncertainties about its significance and board impact, I would like to maintain a conservative score.

---

> > ### Author Response · Authors · 2024-08-31
> > **Response to DMjY**
> >
> > Dear reviewer, thank you for your response. On your concern over JaxMARL's significance, we would make two points. First, we'd like to point out that we believe our work is a significant improvement over the current available MARL frameworks. JaxMARL features high-quality implementations of a wide range of algorithms on a wide range of environments, all at significantly improved speeds. The single-file style, popularised by CleanRL [1], along with the breadth of environments and algorithms and notable speed, allow researchers to iterate faster and explore new research avenues. Your objections on SMAX regard a significant, but small part of our overall project.
> >
> > Secondly, you claim that you are "uncertain whether SMAX will be widely accepted in the MARL community". This is a contentless objection. It is always uncertain whether a benchmark will be accepted when it is released. SMAC is not a perfect, unchangeable evaluation tool. We, along with previous work [2], identify and address serious issues with SMAC and propose fixes in this paper. There is clearly value in this. While we acknowledge the collective action problem of changing evaluation standards, we believe that the benefits of our library will encourage people to use our code and therefore evaluate on this improved environment. The work should be assessed on the value it brings by correcting issues with SMAC and SMACv2.
> >
> > ---
> >
> > [1] Huang, Shengyi, et al. "Cleanrl: High-quality single-file implementations of deep reinforcement learning algorithms." Journal of Machine Learning Research 23.274 (2022): 1-18.
> > [2] Ellis, Benjamin, et al. "Smacv2: An improved benchmark for cooperative multi-agent reinforcement learning." Advances in Neural Information Processing Systems 36 (2024).

---

### Official Review · Reviewer_Vsti · 2024-07-25
**Good contribution for the MARL community**

**Rating:** 8
**Confidence:** 4

**Review:**

- Quality: the paper provides detailed methodology, thorough experiments, and significant benchmarking results.
- Clarity:  the structure and presentation are well-organized, making the complex topics accessible.
- Originality: JaxMARL is the first to leverage JAX for GPU-accelerated MARL environments and algorithms, introducing novel environments and benchmarking protocols.
- Significance: addresses critical bottlenecks in MARL research by drastically improving computational efficiency and providing robust evaluation tools.

Pros:
- Innovative approach: First open-source library combining GPU acceleration with MARL environments and algorithms using JAX.
- Significant speedup: Demonstrates up to 12500x speed improvements, enabling rapid experimentation and thorough evaluations.
- Comprehensive benchmarking: Provides extensive benchmarks and evaluation recommendations.
- New Environments: Introduces SMAX and STORM, enhancing the diversity and complexity of available MARL environments.
- User-Friendly: Ensures ease of use with Python-based implementations and a clear interface inspired by existing libraries.

Cons:
- Specific Focus: Primarily focuses on cooperative and general-sum game settings, which may limit its immediate applicability to other MARL scenarios.
- Limited to extending popular MARL environments

**Strengths:**

- Significance of the contribution: JaxMARL introduces the first open-source library combining GPU acceleration with MARL environments and algorithms, addressing critical computational bottlenecks and enabling faster, more efficient MARL experimentations.
- Relevance to research community: By providing JAX-based implementations of popular MARL environments and algorithms, JaxMARL is highly relevant to MARL researcher, facilitating rapid experimentation and thorough evaluations.
- Quality of the research: The paper is well-structured, with detailed methodology and extensive benchmarking, demonstrating significant speed improvements and validating the effectiveness of JaxMARL across diverse MARL scenarios.

**Additional Feedback:**

Typo:
-	Page 6: “Furthermore, in Figure 2 we find that PPO is 6 times faster” -> “Furthermore, in Table 2 we find that PPO is 6 times faster”

**Clarity:**

The article is very-well written and structured. It easily reads, providing a clear overview and motivation of the contributions, without going in useless details in the main paper. All information is however accessible via the supplementary material and/or the Github repository

**Correctness:**

Experiments of the JaxMARL library are properly conducted. Both performance and speed comparison are analyzed. Section 5.3 is also dedicated to assessing the “Algorithm and Environment Correctness”.

**Documentation:**

A dedicated Github repository is provided, which contains comprehensive information: the code, information on the provided environments and algorithms, some quick start guide, installation instructions, a Dockerfile and pointers to some reference libraries. All necessary information to reproduce the results is provided.

**Ethics:**

No ethical concerns.

**Limitations:**

Limitations are clearly listed in the Conclusion section.

**Opportunities For Improvement:**

- While JaxMARL provides significant computational improvements, its primary focus on cooperative and general-sum game settings may limit its applicability to other MARL scenarios, such as competitive environments.
- Despite thorough benchmarking, the observed speedups are less pronounced for off-policy, value-based methods, indicating that further optimization is needed to fully leverage JAX's potential in all MARL contexts.
- As mentioned by the authors, handling environments with variable numbers of agents or massive observation sizes remains challenging, and the current implementations may require significant customization for specific research needs, limiting the tool's immediate versatility and robustness across all possible MARL applications.

**Relation To Prior Work:**

The related work on both MARL libraries and algorithms and on HW accelerated and JAX-based libraries are comprehensive and up-to-date. These clearly outline the novelty of the prposed JaxMARL library.

**Summary And Contributions:**

The authors introduce JaxMARL, an open-source, Python-based library combining GPU-enabled efficiency with multi-agent reinforcement learning (MARL) environments and algorithms. Leveraging JAX, it significantly accelerates MARL training, showing a 14x to 12500x speed improvement over traditional approaches. It includes JAX implementations of popular MARL environments and introduces two new suites, SMAX and STORM.

Contributions:
- JAX Implementations: Fast, GPU-accelerated versions of popular MARL environments.
- New GPU accelerated MARL Environment Suites: SMAX (StarCraft Multi-Agent Challenge) and STORM for enhanced MARL research.
- Popular Algorithms in JAX: Implementations of IPPO, MAPPO, and QMIX with significant speed improvements.
- Comprehensive Benchmarking: Thorough benchmarking of speed and correctness, comparing favorably to existing CPU-based implementations.
- Evaluation Recommendations: Best practices and scripts for large-scale evaluation and plotting in MARL research.

---

> ### Author Rebuttal · Authors · 2024-08-18
>
> Thank you for your positive review, particularly for highlighting the significance of our contribution and the quality of our research. Please find our responses below.
>
> ## Specific Focus of Environments
>
> While some of the JaxMARL environments are only for cooperative or general-sum settings, several also apply to other MARL scenarios:
> - STORM enables any matrix game to be represented as a grid-world scenario, allowing for general-sum, competitive or mixed incentive settings.
> - MPE contains twelve scenarios, while some are cooperative (e.g. Simple Spread and Simple Reference), the majority are competitive (e.g. Simple Tag, Simple Crypto and Simple Push).
> - SMAX supports competitive self-play.
>
>
> ## Speed of Off-Policy, Value-Based Methods
>
> We agree that the observed speedups are less pronounced for off-policy, value-based methods. As mentioned in our global response recent research [1] has leveraged JaxMARL to address this.
>
> ## Versatility and robustness across all possible MARL applications
>
> * Massive observation sizes are only challenging for the current Q-Learning approaches, which recent research [1] has addressed.
> * The number of agents within an episode can be varied; however, due to JAX's JIT compilation, the *maximum number* possible of agents within an episode must be specified when the environment is instantiated.
> * We raise the customisability of JaxMARL in our global response.
>
>
> ## Conclusion
> Thank you again for your helpful review, we hope we have addressed your comments satisfactorily, and welcome further discussion.
>
> ---
> [1] Gallici, Fellows, et al. "Simplifying Deep Temporal Difference Learning." arXiv preprint arXiv:2407.04811 (2024).

---

> > ### Comment · Reviewer_Vsti · 2024-08-27
> > **Satisfactory answers**
> >
> > Dear authors,
> >
> > Thank you for the general and specific answers. These are well motivated and I do not have any further comments or questions.

---

### Official Review · Reviewer_Zxvy · 2024-08-04
**Review of JaxMARL, a library for end-to-end MARL on JAX**

**Rating:** 6
**Confidence:** 4
**Clarity:** Yes.

**Review:**

The paper is well-written and constructed. The proposed library provides an end-to-end framework for MARL domains with fast training time on accelerator devices like GPUs. It completes an important piece of the puzzle in the end-to-end RL field. Here is a summary of the pros and cons of the paper. The details are listed in the **Strengths, Opportunities For Improvement, and Limitations** sections below.

pros:

- Provide the first JAX-based MARL environment suites, especially a sophisticated SMAX environment.
- Provide recommended minimal environment evaluation sets for researchers under different interest settings
- The paper is well-written and constructed with comprehensive experimental results.

cons:

- Lacks comprehensive benchmarks for the correctness\equivalence of some environments.
- re-implement of existing environments. Limited novelty from the new library.
- Some experiments do not clearly explain how they were conducted.

**Strengths:**

- This is the “first” library for JAX-based MARL environments if the PGX library (including 2-player games such as go and chess) is not considered. Besides, two typical types of MARL algorithms are implemented and tested as baselines for future research.
- Impressive speed-up for MARL: I believe this library will complement the JAX-based end-to-end RL community and help further research of algorithms under cheap evaluation time that sample efficiency is not strictly considered, such as on-policy RL algorithms, population-based RL, and Evolutionary Computation.

**Additional Feedback:**

None

**Correctness:**

There are some potential concerns about the evaluation statement as mentioned in **Opportunities For Improvement**.

**Documentation:**

Yes, the proposed library is well documented.

**Limitations:**

- The implemented environments are all from existing environments. So there are still limited choice to test the MARL algorithms.
- Under the un-batched env case, the speed may not be better than the existing CPU-base implementation, as shown in Table 3.
- The novelty may be limited. I cannot get new insights from the paper. For example, there is no discussion about the balance of training and sampling for MARL algorithms in this new pattern when sampling time is cheaper compared to traditional settings, i.e., how many sampling timesteps are for one training update.

**Opportunities For Improvement:**

- the details of how the speed-up is calculated and whether it is conducted under fair conditions are not clear.
    - How did the authors test the results of Figure 5a and Figure 5b? Since the library both implements the environments and baselines, did they test on the same JaxMARL environments with different implementations of algorithms in JaxMARL and PyMARL?
    - How was the speed-up tested in the statement of “… and 14x quicker on MPE for a single run, and 12,500x for vectorized training runs”? Did they test on the same JaxMARL environments with different algorithm libraries? And what was the termination condition when comparing the training time? Did they use the total sampled timesteps and the same rollout timesteps and training batch size? I did not find the details in the full paper and appendix.
- As mentioned in the paper, some environments are identical to the original environments, while SMAX, STORM have different implementations. There should be more comparisons between the new JAX-based environment and the original counterpart in Sec 5.3.
    - Specifically, due to JIT optimization and the implementation of accelerator math library (e.g., Nvidia cuBLAS), along with the backend difference in MABrax, there will be some precision discrepancy for these “identical” environments, which could potentially cause huge performance differences for MARL algorithms. It will be better to tell the readers by experiments that if they can directly replace the original environments and expect the same performance results.

**Relation To Prior Work:**

Yes. The manuscript clearly distinguishes it from prior works.

**Summary And Contributions:**

The manuscript proposes a new library that provides JAX-based MARL environments and baseline algorithms. The implemented environments help the development of end-to-end MARL training, which massively improves the training speed due to the utilization of accelerator devices like GPUs, vectorization, and JIT techniques of JAX. Moreover, the library provides two types (PPO and Q-learning) of MARL algorithms as baselines, and the experiments demonstrate the speed-up of this end-to-end training routine. The paper additionally examines these JAX re-implemented environments with corresponding existing environments.

---

> ### Author Rebuttal · Authors · 2024-08-18
>
> Dear Reviewer, thank you for your thorough review and constructive comments! We appreciate you mentioning that our paper "is well-written" with "comprehensive experimental results" and that JaxMARL achieves an "impressive speed-up for MARL". Please find our response to your comments below:
>
> ## Environment Correctness and Novelty
>
> >Lacks comprehensive benchmarks for the correctness\equivalence of some environments.
>
> In Section 5.3, we set out correctness results for each environment where applicable, with Coin Game, STORM and Switch Riddle omitted for the following reasons:
> * For Coin Game, we were unable to find an existing implementation to compare against.
> * STORM is new and unlike SMAX has no applicable environment for comparison.
> * Switch Riddle has a simple unit test against a NumPy implementation, housed in our repository's test scripts. However, as it is primarily a debugging tool, we did not feature results for this environment in the paper.
>
> > ...did they test on the same JaxMARL environments with different implementations of algorithms in JaxMARL and PyMARL?
>
> As we have separately demonstrated the correctness of our environment implementations where applicable, we have not run JaxMARL environments with baselines from different libraries.
>
>
> > re-implement of existing environments. Limited novelty from the new library.
>
> In our global response, we outline how JaxMARL still represents a significantly novel contribution.
>
> > As mentioned in the paper, some environments are identical to the original environments...
>
> SMAX and STORM are new environments not different implementations of existing ones. While they were inspired by SMAC and Melting Pot 2.0 respectively, our Appendix outlines in detail how the improvements SMAX and STORM make mean they are not comparable to the environments which inspired them.
>
> As MABrax uses Brax as its backend, not MuJoCo, results should not be directly compared with MaMuJoCo.
>
>
> While JAX's JIT compilation can cause very minor differences to a function's numeric outputs due to the XLA compiler making algebraic simplifications (see the FAQ section of the JAX online docs for details: https://jax.readthedocs.io/en/latest/faq.html#jit-changes-the-exact-numerics-of-outputs), these differences are negligible and they do not result in changes to an environment's function.
>
> To illustrate this, we have repeated our MPE transition matching test described in Section 5.3 with a tolerance of $1\times 10^{-10}$ rather than the $1\times 10^{-4}$ used previously. Using a uniform-random policy we rollout 1000 episodes and ensure that at each transition, all observations and rewards are within this tolerance. As such, researchers can directly replace the original environments and expect the same performance results.
>
>
> ## Experiment Details
>
> > Some experiments do not clearly explain how they were conducted.
>
> We apologise for our oversight in omitting full details for the experiments you have mentioned, we have provided these below and updated our manuscript to include full descriptions.
>
> > How did the authors test the results of Figure 5a and Figure 5b?
>
> Thank you for spotting that the hardware details for this test were missed in the Appendix. Results for Figures 5a & 5b were collected on the same hardware as used for Table 3 - a single NVIDIA 255 A100 GPU and AMD EPYC 7763 64-core processor.
>
> > How was the speed-up tested in the statement of “… and 14x quicker on MPE for a single run, and 12,500x for vectorized training runs”?...
>
> These tests compare the speed of JaxMARL's training pipeline to those of two popular MARL libraries, `PyMARL` and `On-Policy`. As we compare training pipelines, JaxMARL trains using JaxMARL's algorithm and environment implementations while the others use their algorithm implementations with the compatible CPU-based environment implementations - SMAC and PettingZoo's MPE.
>
> Thank you for raising that the exact details for these tests were omitted from the manuscript. For each test, all libraries used identical hyperparameters and environment scenarios, 2s3z for SMAC/SMAX and Simple Spread for MPE. The hyperparameters used are listed in Section 7 of our Appendix. For training, the termination condition was for all PPO update steps to have occurred.
>
>
> ## Limitations
> > The implemented environments are all from existing environments.
>
> Please see our global response for details about this.
>
> > un-batched env case
>
> As reported in Table 3, this is true for several environments but it should be noted that during RL training it is very rare to use the un-batched case. As such, this limitation will not affect JaxMARL's impact for researchers.
>
> > The novelty may be limited...
>
> We are not entirely sure what the reviewer means by this. We include details of our hyperparameters in the Appendix as well as a discussion of the advantages of Jax in our background section.
>
> We would also like to raise that detailed performance breakdowns and comparisons to the previous paradigm are challenging. For example, one significant cost of running environments on the CPU is that you must ship data between CPU and GPU memory. However, it is unclear whether this should be ascribed to rollout or update time. Similarly, rollout time versus update time will depend on rollout length and the memory performance of that. Even if we performed such experiments, which we leave to future work, it is not clear that we would get widely applicable answers.
>
> ## Conclusion
> Thank you again for your thorough review. We hope that the reviewer feels we have addressed their questions and welcome any further discussion. We also ask that, if all their concerns are met, the reviewer consider increasing their support for our paper.
>
> ---
> [1] Gorsane, et al. "Towards a Standardised Performance Evaluation Protocol for Cooperative MARL." Advances in Neural Information Processing Systems 35 (2022): 5510-5521.

---

> > ### Author Response · Authors · 2024-08-28
> > **Response to Zxvy**
> >
> > Dear Reviewer, we hope you have had time to review our response. With the discussion period nearing its end, please could you let us know whether we have addressed your concerns or if you still have any questions.

---

> > ### Comment · Reviewer_Zxvy · 2024-08-30
> > **Fewer remaining concerns**
> >
> > **Lacks comprehensive benchmarks for the correctness\equivalence of some environments**:
> >
> > Although Sec. 5.3 and Fig. 6 discuss several environments, but I think they are not adequate. Since the proposed number of environments is a manageable size, it is advised to test all applicable environments with their original counterpart under the same baseline algorithm in JaxMARL and plot the training curves (which could be placed in the appendix). It should also include MABrax, as mentioned in the second entry of "Opportunities For Improvement." It will give readers a clear overview of how many returns they could expect and whether their previous experience with the method remains useful.
> >
> > **Speedup in environments**:
> >
> > Since SMAX has simplified the running logic of SMAC, when comparing the training speed, the 31x speedup is computed in an unfair way. The value should only be computed in identical environments.
> >
> > **Novelty issue**:
> >
> > Benefiting from JAX, the environments could be massively paralleled (e.g., 10k #envs) with great speedup, which is the main contribution of the framework. However, the contribution is not deemed as complete novelty. Compared to previous framework, how the proposed baselines handle the incoming massive rollout data needs to be extensively discussed. For example, there might be new trade-off between training updates and the number of sampled timesteps/episodes controlled by hyperparameters like NUM_ENVS and BATCH_SIZE. How to tune these hyperparameters to let the algorithms effectively learn from the highly paralleled environments needs further discussion, especially in the off-policy Q-learning settings. Solving these concerns with the baselines will help the community understand how to effectively utilize and take advantage of the framework, and give some new insights for future development.
> >
> > Based on the above concerns, along with some writing issues, the paper still has the opportunity to improve, so I decided to maintain my rating.

---

> > > ### Author Response · Authors · 2024-08-31
> > > **Response to Zxvy**
> > >
> > > Dear Reviewer, thank you for your response and for outlining your concerns.
> > >
> > > ## Equivalence of Environments
> > >
> > > As the reviewer notes, we have taken care to evaluate the similarity of our environments to the originals where possible. How we have done this is extensively detailed in our paper. Where we have not, it is for good reason. We have extensively benchmarked SMAX, MPE, Overcooked and Hanabi to compare them to their prior versions. We have not done so for MABrax because the different dynamics make comparing results challenging. Additionally, we could not find results in the literature that trained PPO on MAMuJoCo as most papers seemed to use MAA2C or other approaches. It is therefore not clear that there is a suitable baseline with which to compare. Finally, we would like to note the extensive benchmarking which we *have* done and its value to the community.
> > >
> > > ## Speedup in Environments
> > >
> > > We would like to point out that **SMAX is simplified only in that it does not run the SC2 game engine**. The actual dynamics of the game are extremely similar, with the differences resulting from uncertainty about how the SC2 game engine is implemented, or rare/minor aspects of SC2. This is backed up by the similar overall performance of algorithms in SMAX compared to SMAC.
> > >
> > > Secondly, we point out the differences between SMAX and SMAC in the main text and expand upon them extensively in the Appendix. We are not aiming to claim a higher performance number than reasonable by comparing two different environments, but to point out performance benefits from a real use case. The speedup from SMAX compared with SMAC is the speedup that you can hope for if you can re-implement a complicated environment in JAX. While environments such as MPE are useful points of comparison, their CPU versions are already fairly simple and cannot benefit from JAX's parallelism as much as SMAX for example.
> > >
> > > ## Novelty Issues
> > >
> > > We agree that JAX-based RL has very different properties compared to previous CPU-based environments and that this is particularly relevant for Q-learning. In fact, and as mentioned in our global response, this observation inspired a significant project led by one of the lead authors of our paper [1] to produce PQN, a Q-learning algorithm suited for use with Jax-based environments. Given the complexity of the discussion there, we believe that these interesting questions about how best to adapt Q-learning and other methods for JAX-based environments are best left for future work.
> > >
> > > ---
> > > [1] Gallici, Matteo, et al. "Simplifying Deep Temporal Difference Learning." arXiv preprint arXiv:2407.04811 (2024).

---

### Author Rebuttal · Authors · 2024-08-18

$\newcommand{Zxvy}{\textcolor{blue}{\mathrm{Zxvy}}}$
$\newcommand{Vsti}{\textcolor{red}{\mathrm{Vsti}}}$
$\newcommand{DMjY}{\textcolor{orange}{\mathrm{DMjY}}}$
$\newcommand{Vpt}{\textcolor{green}{\mathrm{Vp7t}}}$

We would like to thank all the reviewers for the time taken to review our submission and for your constructive comments. Below, we address the common points raised by several reviewers. For clarity, reviewers are colour-coded as follows:

- $\textcolor{blue}{\mathrm{Blue -}} \Zxvy$
- $\textcolor{red}{\mathrm{Red -}} \Vsti$
- $\textcolor{orange}{\mathrm{Orange -}} \DMjY$
- $\textcolor{green}{\mathrm{Green -}} \Vpt$

## Similarity of SMAX and SMAC

Reviewers $\Zxvy$ and $\DMjY$ ask questions on the similarity of SMAX to SMAC. **SMAX is not a reimplementation of SMAC, nor is it an easier version.** SMAX is simplified in that running the environment does not require running the full game of StarCraft II. However, we made significant efforts to keep the dynamics of SMAC and SMAX as similar as possible, while not tying ourselves unnecessarily to the original values.

**As well as running faster, SMAX also addresses issues present in both SMAC and SMACv2.** For example:

* In SMAC and SMAX, the reward has two components: one for winning and one for depleting the enemies' health. However, in SMAC the component for depleting enemy health scales with the number of agents. As a result, there is a discrepancy in reward attribution between scenarios with different numbers of agents. In SMAX this is rectified as while the reward signal retains these same components, the ratio between them is fixed as explained in our Appendix.
* SMAC's heuristic (enemy) policy has several flaws in both SMAC and SMACv2. The specific flaws and how SMAX fixes them are detailed in our Appendix.
* Because SMAX does not rely on the StarCraft II game engine, it is much more customisable, allowing scenarios with units from different races, as well as training through self-play. Furthermore, as SMAX is entirely written in Python, its dynamics can be easily changed to evaluate methods as needed.


## Environments being reimplementations

Multiple reviewers ($\Zxvy$, $\Vsti$, $\Vpt$) raise that many of JaxMARL's environments are reimplementations of existing ones. While this is true, JaxMARL brings significant novelty as:
* We provide two new environments, SMAX and STORM.
* Our reimplementations enable significant speedups, due to the JAX framework. This has the potential to significantly accelerate current research and open up new research avenues which previously would have been compute-constrained.
* JaxMARL's combination of environments exist under one API for the first time. This allows researchers to easily evaluate methods across multiple popular environments, allowing for more thorough evaluation.

The importance of these contributions is emphasised by previous work [1] establishing that many MARL works only evaluate on one, or a small number of environments. This causes methods to overfit to a small number of environments (primarily SMAC and MPE). As JaxMARL provides fast implementations of a wide range of environments within a single easy-to-use repository, we hope we will help alleviate the evaluation issues within the field as reported by [1].


## Speed-up for Q-Learning Methods

$\Vsti$ and $\Vpt$ note that the speed-up for QLearning algorithms is not as significant as for PPO. We would like to highlight that PPO is based on parallelised exploration and thus benefits directly from environment vectorisation. In contrast, our QLearning baselines are based on PyMARL, which uses the traditional DQN pipeline that interacts with one environment at a time, making it harder to fully leverage environment vectorisation. We chose these pipelines for compatibility with previous MARL research. However, new research is already leveraging JaxMARL to create more efficient QLearning baselines which fully utilise parallelisation, outperforming even MAPPO. See for example the PQN version of VDN [2].

## Environment and Algorithm Customisation

Reviewers $\Vsti$ and $\Vpt$ raise questions on customising JaxMARL. Due to our algorithm implementation philosophy and the functional nature of JAX, we believe JaxMARL is readily customisable:
* Inspired by CleanRL, JaxMARL features single-file implementations for all baselines. This ensures that they are easy to customise for a specific research need.
* Due to JAX's functional nature, we have found that many of our environment implementations are often more understandable, and less prone to side effects when customising than current Python implementations. As such, we believe our environments are readily customisable for specific research questions.

---
[1] Gorsane, et al. "Towards a Standardised Performance Evaluation Protocol for Cooperative MARL." Advances in Neural Information Processing Systems 35 (2022): 5510-5521.

[2] Gallici, Fellows, et al. "Simplifying Deep Temporal Difference Learning." arXiv preprint arXiv:2407.04811 (2024).

---

### Decision · Program_Chairs · 2024-09-26

**Decision:**

Accept (Poster)

**Comment:**

This paper presents a JAX-based framework that accelerates multi-agent reinforcement learning (MARL) training with environments and algorithms designed for efficient use of hardware acceleration. The authors introduce the JaxMARL library, which provides support for a variety of well-established MARL environments alongside two new environments, SMAX and STORM. The framework demonstrates significant computational speedups, with claims of up to 12,500x in some cases, allowing for faster experimentation and evaluation in MARL research.

Strengths:
- The library leverages JAX’s capabilities for vectorization and GPU acceleration, resulting in substantial speed improvements, which allow for rapid and scalable training across multiple MARL environments. Reviewers were particularly impressed by the acceleration, citing the potential for significant impact on the research community.
- JaxMARL offers implementations of several popular MARL environments, such as SMAC, Hanabi, and MPE, while introducing two new environments, SMAX and STORM. The breadth of environments makes it a valuable benchmarking tool for MARL researchers, addressing common issues of overfitting to a small number of environments.
- Reviewers commended the quality of the implementation, which features single-file, cleanly written code inspired by CleanRL. This makes it easier for researchers to modify and customize the library for their needs.
- JaxMARL was recognized as an important tool that could alleviate current evaluation challenges in MARL research. By providing a fast, accessible suite of environments, the submission contributes to raising the standards of empirical evaluation in the field.

Weaknesses:
- While JaxMARL brings technical improvements, reviewers expressed concerns about the limited novelty in terms of new environments or algorithms. Most of the provided environments are re-implementations of existing ones, with SMAX and STORM seen as extensions rather than fundamentally new concepts. This affects its perceived contribution to advancing the field.
- Some reviewers requested more detailed comparisons between SMAX and SMAC to determine whether the insights gained in SMAX can be transferred to SMAC. Although the authors addressed this in their rebuttal, concerns persisted about how these environments differ and whether SMAX offers significant advantages.
- While the library offers substantial speedups, some reviewers noted that the flexibility of the environments is somewhat limited. For instance, the number of agents cannot vary dynamically due to JAX’s JIT compilation. This restricts the use of the library for certain types of research, such as dynamic policy selection or league training, seen in complex MARL systems like AlphaStar.
- Although the framework demonstrates excellent speedups for on-policy algorithms like PPO, the speed improvements for off-policy algorithms like Q-learning are less pronounced. This suggests room for further optimization in this area.

Overall Assessment: The reviewers agreed that JaxMARL makes a strong technical contribution, with clear practical benefits to the MARL research community. The computational speedups were universally praised, as they address a major bottleneck in current research.